# Unsupervised Learning Methods for Data-Driven Vibration-Based Structural Health Monitoring: A Review

**DOI:** 10.3390/s23063290

**Published:** 2023-03-20

**Authors:** Kareem Eltouny, Mohamed Gomaa, Xiao Liang

**Affiliations:** Department of Civil, Structural and Environmental Engineering, University at Buffalo, The State University of New York, Buffalo, NY 14260, USA

**Keywords:** machine learning, deep learning, structural health monitoring, damage detection, unsupervised learning, novelty detection, anomaly detection, outlier analysis, vibration-based methods, neural networks

## Abstract

Structural damage detection using unsupervised learning methods has been a trending topic in the structural health monitoring (SHM) research community during the past decades. In the context of SHM, unsupervised learning methods rely only on data acquired from intact structures for training the statistical models. Consequently, they are often seen as more practical than their supervised counterpart in implementing an early-warning damage detection system in civil structures. In this article, we review publications on data-driven structural health monitoring from the last decade that relies on unsupervised learning methods with a focus on real-world application and practicality. Novelty detection using vibration data is by far the most common approach for unsupervised learning SHM and is, therefore, given more attention in this article. Following a brief introduction, we present the state-of-the-art studies in unsupervised-learning SHM, categorized by the types of used machine-learning methods. We then examine the benchmarks that are commonly used to validate unsupervised-learning SHM methods. We also discuss the main challenges and limitations in the existing literature that make it difficult to translate SHM methods from research to practical applications. Accordingly, we outline the current knowledge gaps and provide recommendations for future directions to assist researchers in developing more reliable SHM methods.

## 1. Introduction

### 1.1. Background

The integrity of civil structures gradually decreases due to use and operational conditions while also being at risk of unforeseeable hazards such as seismic events. To avoid life and capital losses due to sudden and long-term damage, techniques and standards for structural inspection were put forward, such as visual inspection and nondestructive evaluation techniques. These traditional structural inspection techniques, however, can be expensive, time-consuming, and unsafe for human workers. Structural health monitoring (SHM) attracted the attention of researchers in the past decades due to its ability to provide real-time structural condition assessment and the progress made in hardware development [1,2,3].

Farrar et al. [4] have put forward a statistical pattern recognition paradigm that split the SHM framework into four processes: operational evaluation, data acquisition, feature selection, and statistical model development for feature discrimination. While articles discussed in this review primarily focus on the latter two processes, we find that more studies give more attention to feature selection than model development. The statistical models attempt to identify the existence, location, and severity of damage [5]. There has been a steady development of SHM methods in the past decades that are vibration-based and vision-based. Vision-based SHM benefits from the advances in computer vision technology and unmanned aerial systems to provide autonomous visual inspections [6,7,8]. Vibration-based SHM uses recorded vibration data of the structures to identify structural damage which could be difficult to identify via visual inspections.

Two major types of vibration-based SHM are available: model-based and data-driven. Model-based techniques involve system identification and model updating techniques [9]. They rely on expert knowledge to build accurate, physics-based models of structures that are calibrated based on real structure measurements [10,11,12,13]. However, model updating techniques can be expensive, time-consuming, and prone to modeling errors, especially for complex structures [14,15].

Data-driven SHM uses either supervised or unsupervised learning methods to train a statistical model to identify damage and faults based on engineered damage-sensitive features. In supervised learning, the training data are labeled and include all structural states including undamaged and damaged conditions. Examples of supervised SHM methods include support vector machine (SVM) [16,17], decision trees [18], neural networks [19,20], and deep neural networks [21,22,23,24].

While supervised learning can be ideal for vision-based SHM, as training data are relatively attainable and transfer learning is feasible [25,26,27], the practicality of these methods are debated for vibration-based SHM [2]. The acquisition of the damage condition data needed for supervised learning is challenging. One possible way is by relying on a physics-based model, which can be nontrivial for complex structures. Another way is to acquire it from laboratory or field experiments, and this also can be impractical for most structures. Additionally, transfer learning research for SHM has not yet matured, as structures tend to have different properties and site conditions even if the structures themselves are similar.

Unsupervised learning can offer a more practical alternative to supervised learning for SHM systems. The bulk of unsupervised learning methods for vibration-based SHM are novelty detection methods. In the context of SHM, the damaged state data is not needed for training in an unsupervised learning setting. Instead, models can be trained using data from the normal condition only, which are often available in abundance. Additionally, unlike supervised learning, which trains models to detect damage types that are only considered in the dataset, unsupervised learning may detect any system change that can be picked up by the model. However, unsupervised learning is generally less accurate in localizing and quantifying damage compared to supervised learning [28].

A flowchart of an example unsupervised learning-based damage detection algorithm is shown in Figure 1. After data acquisition and cleansing, the sensor signals pass through three stages. First, the signal readings may undergo preprocessing to facilitate feature extraction, such as data normalization. Second, damage-sensitive features are extracted from the input data using dimensionality reduction, signal decomposition, or other techniques. In some methods, multiple layers of feature extraction are proposed. For instance, some methods transfer the signals into the frequency domain and then reduce the dimensions using a machine-learning model. Third, a statistical inference process is proposed involving the estimation of one or multiple damage indicators and a form of a statistical test (e.g., hypothesis testing) to diagnose damages. The last decision-making part is occasionally absent from the framework, and the users would need to manually inspect the damage indicators for health assessment. Unsupervised learning-based SHM methods require establishing a reference (e.g., thresholding) and/or training the model parameters (machine learning models) using the undamaged structure data in order to assess structural health during operation.

A clear example of this process can be seen in the unsupervised learning damage detection methods introduced by Worden et al. [29,30]. Their feature extraction process involves signal processing methods (e.g., transmissibility), while multivariate outlier analysis and the Mahalanobis squared distance (MSD), as a discordancy measure, were used for decision-making. The basic principle in outlier analysis, or novelty detection in general, is that after fitting a distribution to the normal condition data, observations that fail the outlier test are labeled novel or damaged. They also introduced a threshold estimation technique based on a Monte Carlo simulation, which takes into account the training data size. This damage detection framework would inspire the development of many subsequent novelty detection-based SHM systems. For example, Gul and Catbas [31] presented an MSD-based outlier analysis method for damage detection combined with features extracted from autoregressive (AR) models.

Artificial neural networks were also used for unsupervised-learning SHM. Sohn et al. [32] presented an auto-associative neural network (AANN) for damage detection using features extracted from time series models showing good results under environmental variations. In later studies, damage detection methods based on self-organizing neural networks [33] and wavelet neural networks [34] were also introduced. While some novelty detection methods can be considered as one-class classifiers, cluster analysis techniques, such as k-means clustering [35] and fuzzy c-means [36], grouped the data in multiple clusters.

### 1.2. Related Work

Multiple studies have reviewed vibration-based SHM methods in the past, both supervised and unsupervised learning. Doebling et al. [37] and Sohn et al. [38] jointly reviewed methods that were developed before 2001. Carden et al. reviewed vibration-based SHM research from 1996 to 2003 [39]. Fan and Qiao [40] provided a comprehensive review on damage detection methods using modal parameters. Many others have focused on reviewing signal processing techniques for SHM [41,42,43,44,45].

More recently, some studies have conducted updated literature surveys on SHM research. Hou and Xia [46] reviewed vibration-based damage identification techniques introduced from 2010 to 2019, including methods that are based on Bayesian learning. Sun et al. [47] and Zinno et al. [48] have reviewed SHM methods for bridges. Gharehbaghi et al. [49] performed a critical review on different types of SHM methods and also highlighted some commonly-used SHM benchmarks. Gordan et al. [50] provided a comprehensive review on data mining techniques in SHM from classical to state-of-the-art methods. Cawley [51] discussed the potential reasons for delaying the application of SHM in the industry in contrast with machine condition monitoring.

The rapidly advancing research in sensor technology and machine learning (ML) has positively impacted civil engineering research in multiple fields [52]. SHM research was no exception, as studies in recent years show an increasing number of SHM methods that rely on deep learning (DL) architectures (Figure 2) [53,54]. Flah et al. [55] summarized ML-based SHM methods including DL and reinforcement learning applications. Avci et al. [56] also reviewed ML and DL damage detection methods but only focused on vibration-based applications. Sony et al. have conducted a systematic review of SHM methods that are based on convolutional neural networks (CNNs) [57]. While some of these works briefly discuss unsupervised learning techniques as part of the SHM literature survey, there is a lack of review studies dedicated to unsupervised learning SHM methods as the primary focus.

### 1.3. Aim and Methodology

This work aims to review the latest unsupervised learning vibration-based structural health monitoring techniques. We provide a review of 83 unsupervised learning SHM published between 2012 and 2023, which are summarized in Table 1. It should be emphasized that this is not an exhaustive list of all unsupervised learning-based SHM methods in this period, but rather a curated list of peer-reviewed articles that provide an overview of the state-of-the-art. The selection method of these articles can be summarized as follows:Peer-reviewed articles from 2012 to 2023 were selected from well-established academic databases including Web of Science, Science Direct, ASCE Library, Wiley Online Library, IEEE Xplore Digital Library, and Sage.The search was conducted using relevant keywords, including “structural damage detection”, “unsupervised damage detection”, “unsupervised structural health monitoring”, “structural novelty detection”, “anomaly damage detection”, etc.Two rounds of screening were performed.
∘In the first round, the titles, abstracts, and keywords of the articles were checked for relevancy to the search topic.∘A large number of papers were selected for the second round. in which careful reading and analyzing of the complete article was performed.Papers were given relevancy scores based on aspects such as the learning mode, method, objectives, feature types, application (structural or non-structural), the decision-making process, thresholding, and results.

Finally, based on our search methodology, papers that were closely related to the review topic were selected. We also present some of the most commonly-used benchmarks in validating the unsupervised SHM methods. These datasets are selected based on their recurrence in the reviewed list of articles. Furthermore, we discuss a number of the open issues for the practical implementation of unsupervised learning damage detection methods, as well as future research trends. The remainder of the article is organized as follows. The following section describes the popular experimental datasets used in the literature. Section 3, Section 4 and Section 5 provide a detailed review on the papers shown in Table 1, categorized into conventional feature extraction techniques, artificial neural networks, and novelty detection methods based on the key used unsupervised learning technique. Section 6 discusses the challenges and future research in unsupervised learning SHM methods. Finally, a summary and closing remarks are presented in Section 7.

## 2. Popular Datasets for Unsupervised Learning-Based SHM

Lack of training data has always been a significant challenge in the creation of structural damage detection systems. By recognizing the importance of structural health monitoring over the long term, governments, organizations, and researchers have joined forces to provide training data that supports the development of technologies in this field. In this section, we will briefly describe the most common benchmarks that were used for training and testing of unsupervised learning SHM in recent studies.

### 2.1. Z-24 Bridge

The Z-24 bridge is a post-tensioned concrete bridge in which the main girder has a box cross-section with two vents. The middle span length is 30 m, and it has two symmetric 14-m-long bays from each side, as shown in Figure 3. The Swiss bridge, which was constructed in 1963, is by far the most common test benchmark in the reviewed papers. In 1998, the bridge was demolished to allow a new railway to be constructed. During its last year, accelerations and environmental conditions were continuously recorded. These conditions included humidity, rain, wind speed, and directions and temperature. Then several damage scenarios were applied to the bridge and recorded. The bridge dynamics data were collected using 16 accelerometers to record the accelerations at different locations and in different directions. A total of 48 sensors were used to collect the environmental parameters [141]. The bridge was suitable for testing methods’ environmental variability robustness, as the monitored data included the cold months.

### 2.2. Tamar Bridge

Tamar bridge is a major suspension bridge located in Devon, UK to connect Saltash in Cornwall and Plymouth. With an overall length of 643 m, the main span is 335 m while the side spans are 114 m each. The concrete deck is suspended by cables carried by two 73-m-high towers [143]. An upgrade took place in the late 1990s to comply with the new European Union directive to support vehicles up to 40 tons. Acceleration and environmental sensors were installed to record data; however, as all of these observations were taken from the undamaged condition of the bridge, only false positive errors can be identified.

### 2.3. Sydney Harbour Bridge

The Sydney Harbour Bridge (SHB) is a steel through arch bridge in Australia that connects the northern suburbs of Sydney to the city center. The bridge has eight lanes for vehicles and two railway lines, and it is used by many vehicles daily. The structure of the SHB can be divided into three main sections: the southern approach. the northern approach, and the central main span with a length of 503. Lane 7 is made up of an asphalt surface on a concrete deck supported by a combination of concrete and steel jack arches, as shown in Figure 4. Along a total distance of 1.2 km, there are approximately 800 of these jack arches that have been equipped with three tri-axial MEMS accelerometers. One of these sensors is located at the bottom of the jack arch, while the other two are attached to each side (Figure 5). As some of the joints were known to be damaged, the data was collected during operation before and after the repair. If all 2400 sensors were to operate continuously, they would generate approximately 1 TB of data per day. During the recording period, two types of data were collected: event-based data and continuous data. For event-based data, a pre-determined threshold for acceleration was used to trigger the recording after vehicles, usually buses, pass above the sensors, and it lasts for 2 s. The continuous recording started only in 2015, when a 10-min continuous window of recording was stored at five predetermined times in the day [112].

### 2.4. S101 Bridge

The S101 bridge, which crossed the A1 Westautobahn national highway in Reibersdorf in Austria (Figure 6), was a 6.6-m-wide prestressed concrete bridge. The bridge was built in 1960 and eventually demolished due to structural issues and to make room for additional lanes on the highway below. Demolishing the bridge provided an opportunity to conduct tests on the progression of structural damage, and fifteen tri-axial sensors were mounted on the bridge deck to record dynamic responses. The data was continuously recorded from 10 to 13 December 2008, with a sampling frequency of 500 Hz, resulting in 714 data sets, each containing 165,000 samples. The bridge was closed to traffic during the damage testing, so the main sources of excitation were the wind and vibrations from the highway below the deck [144].

### 2.5. Qatar University Grandstand Simulator

The Qatar University grandstand simulator (QUGS) (Figure 7) is a steel frame supported by 4 columns and made up of eight girders, 4.6 m long, and 25 secondary beams, while the cantilevers are 1 m long and the others are 77 cm [146]. The simulator has 30 accelerometers attached to the joints. While a shaker was used to simulate the excitation on the structure, 31 scenarios were conducted to gain the training data. Two scenarios represented the undamaged case and the other 29 scenarios represented different cases of joint damage. Dynamic responses were collected for 256 s at a sampling frequency of 1024 Hz [147]. Due to its dense grid-like sensor layout, the dataset attracted multiple researchers seeking to validate their deep learning SHM methods in recent years [61,69,75].

### 2.6. Los Alamos National Laboratory Three-Story Frame Structure

The Los Alamos National Laboratory (LANL) three-story frame structure consists of four aluminum plates and four aluminum columns, with the plates only permitted to move in the x- direction (Figure 8). Additionally, a central column that is connected to the top plate is added to simulate the damage to the structure as it induces nonlinearity to the structure’s behavior. The structure’s dynamic response is measured using four accelerometers attached to the center of each plate. The testing scenarios can be mainly classified into four main groups. The first one represents the baseline where the structure is considered undamaged. The second one is set to simulate the environmental and operational conditions which are performed by changing the columns’ stiffnesses and masses. The damage is simulated in the third group by using the bumper to introduce the nonlinearities to the structure. Finally, the last group is a combination of the last two groups together to simulate both the damage and the environmental change [148].

### 2.7. IASC-ASCE Benchmark Structure

The IASC-ASCE Benchmark Structure was designed by the IASC-ASCE Structural Health Monitoring Group. It has several numerical simulation studies [149] and was experimentally tested twice [150,151]. It is 3.6 m tall and has two 2.5-m-long spans in both directions (Figure 9). The structure is made up of four floors, and each floor has four steel plates, measuring 1.5 m by 0.65 m, that support the dead load. The frame is made of 300 W steel with S75 × 11 beams and B100 × 9 columns. Each floor also has four 50 mm square steel tubes for in-plane stability and four pairs of 12 mm diameter steel rods for lateral stability. These rods are pretensioned with a torque wrench to ensure consistent force throughout the structure. The structure is equipped with 15 accelerometer sensors, 3 of which are placed on the base and 3 on each floor (north, south, west) of the structure. In addition, there is one temperature and one moisture sensor placed to consider the effects of the environmental effect on the detection process. Various levels of damage were introduced to the structure by removing one or both braces on each floor, resulting in 15 different damage configurations. It is worth noting that testing the mentioned damage scenarios was conducted on four non-consecutive days.

### 2.8. Tianjin Yonghe Bridge

This cable-stayed bridge has two towers, each 55.5 m long, and its main span is 510 m long (Figure 10). After 19 years of the bridge’s operation, cracks up to 2 cm wide were found at the mid-span girder segment. It is suspected that these cracks were caused by vehicles exceeding the weight and volume limits of the bridge’s original design. Additionally, the cables had severely corroded. To adjust these issues, repairs were conducted from 2005 to 2007. During the rehabilitation and repair of the bridge, over 150 sensors on the bridge’s girders, cables, and towers, as well as data acquisition devices in the control room, were installed. There were 14 uniaxial accelerometers permanently attached to the deck and only one biaxial accelerometer installed on the top of one of the towers. To measure wind velocity in all directions and ambient temperature, a temperature sensor and an anemoscope sensor were also placed on the south tower. Additionally, the bridge deck had a weigh-in-motion system installed for all lanes [153].

### 2.9. The Corgo Bridge

The Corgo Bridge, located in the Vila Real District of Portugal, is a long bridge constructed from prestressed concrete box-girders. It is 2796 m in total length and is divided into three parts: the East Sub-Viaduct, the West Sub-Viaduct, and the Central Sub-Viaduct, which is a cable-stayed bridge with a 300-m-long central span (Figure 11). It is held up by a suspension system made up of four semi-fans, each with 22 stay cables. The deck of the Central Sub-Viaduct is 28 m wide and is made of a 3.5-m-high box-girder. It has two carriageways with two traffic lanes each. The pylons of the bridge are about 193 m tall and are directly connected to the deck [96]. The system includes the measurement of various parameters such as bearing displacements, deflections, rotations, forces in the cables, concrete strains, and concrete temperatures. To measure these parameters during operation continuously, both fiber-optic and electric sensors were employed [155].

### 2.10. PI-57 Bridge

The PI-57 bridge is a double-deck structure in France that carries the A1 motorway. It was built in 1965 but encountered issues with cracking and deflection due to insufficient prestressing. To address these problems, the bridge underwent a reinforcement procedure in 2009, which involved adding additional longitudinal prestressing. To determine the efficiency of the procedure and assess the structural behavior under thermal effects, vibration-based monitoring took place. Two campaigns of measurements were carried out before and after the reinforcement, and sixteen piezoelectric accelerometers were used. The tests were performed relying on traffic as the excitation source, and dynamic tests were conducted between October 2009 and April 2010 [128].

## 3. Conventional Feature Extraction Techniques

A summary of SHM methods that primarily rely on conventional feature extraction techniques is presented in this section. The methods are categorized based on the type of feature extraction techniques, which are grouped into two categories: dimensionality reduction methods and signal processing methods. All damage detection methods involve some form of a univariate or multivariate novelty detector, such as outlier analysis, but the methods with more emphasis on the feature extraction part are discussed in this section. Some frameworks also may rely on multiple feature extraction techniques that are performed in series, which is especially the case for low-complexity machine learning methods.

### 3.1. Subspace Analysis-Based Dimensionality Reduction

Raw measurements often contain mixed signals from different sources, each having a different level of contribution. Subspace analysis techniques can be used to provide a set of linear combinations of the signals that best explain the underlying data, resulting in a reduction in dimensions. Principal components analysis (PCA) [156,157] is one of the most common dimensionality reduction techniques in SHM (Figure 12). Singular value decomposition of the normalized data is often used to obtain the principal components, which are the eigenvectors with the highest eigenvalues (variance). While PCA and other linear subspace learning techniques are linear mapping methods, there are nonlinear dimensionality reduction techniques, including PCA variants using the kernel trick (e.g., kernel PCA).

Kesavan and Kiremidjian [139] presented an unsupervised damage detection method based on a hybrid of wavelet transform (WT) and PCA for feature extraction. Using k-means clustering, damage can be hypothesized if more than one cluster is needed to model the features using gap statistics. When compared to features based on time series coefficients [158], the introduced method resulted in more separable observations, persuading the authors to opt for k-means instead of the more complex Gaussian mixture models (GMM) clustering. Additionally, the Euclidean distance between the two clusters can be used as an estimate for damage severity. The method is validated using the numerical simulations of the IASC-ASCE benchmark structure [147]. Owing to the efficiency and practicality of PCA, many researchers preferred using it for dimensionality reduction compared with other methods [67,74,104]. For example, Zhou et al. [111] combined the use of transmissibility and PCA as a way to reduce the number of transmissibility functions by selecting a few components of their projections into the principal components space.

Tibaduiza et al. [126] used PCA mapping to determine four damage indices that can be utilized to detect the occurrence of damage in structural systems. The first two indices are T^2^-statistic and Q-statistic, which can be determined from residual data. Then, the combined index and I^2^ index can be calculated using the two indices. An airplane turbine blade and aircraft skin panel were tested for validation purposes. In addition, aiming for a practical SHM system, Sousa Tomé et al. [105] introduced a damage detection and localization method that uses the residuals from a multilinear regression model of cable forces of cable-stayed bridges as damage-sensitive features. Their novelty detection approach was to use Hotelling T^2^ control charts based on PCA of the model residuals. In their case study of a numerically simulated Corgo Viaduct, their model was able to flag stay cables with area reductions smaller than 1%. Localization was also performed using the relative variation of the T^2^ statistic. In a latter study, however, a multivariate cointegration analysis based on the Johansen test was used, replacing PCA [96].

Moving principal component analysis (MPCA) is a PCA variant that is best suited for continuous condition monitoring [159]. In MPCA, PCA is applied to a sliding window instead of the entire signal to reduce computational costs. Laory et al. [136] studied the impact of using MPCA with four regression methods for structural damage detection. The regression residuals were used as damage indices, which were tested for novelty using a confidence interval of six standard deviations. Upon validation on concrete bridge experimental tests, they concluded that the addition of MPCA improves the damage detection accuracy and increases the computational efficiency. Focusing on real-time bridge monitoring applications, Nie et al. [94] introduced a damage detection and localization method based on fixed MPCA. The method is an improvement over MPCA as the length of the moving window is determined based on the convergent spectrum of cumulative contribution ratio. The principal components’ vectors and eigenvalues are used as damage indices. It was tested on a suspension bridge in Guangdong, China and the method was able to identify the occurrence of a minor non-damaging incident. The study, however, does not provide details on thresholding the damage indices for automated decision making.

Motivated by the limitations of PCA in handling measurement uncertainty and missing data, Ma et al. [77] introduced an anomaly detection method based on probabilistic principal component analysis (PPCA). A probabilistic variant of PCA based on the Gaussian latent variable model, PPCA is generally used when there are missing values in the input data matrix. Two anomaly statistics are used: Q-statistic and T^2^ statistic, while the residual in Q-statistic is used to localize damage. The method was tested in a dataset collected from a revolving auditorium in China, with simulated damage showing high success in identifying damage with and without missing data when compared to traditional PCA. However, damage in members with high redundancy can be challenging under moderate noise conditions.

While PCA and its variants constitute the majority of commonly used dimensionality reduction techniques in SHM, there are other subspace learning methods used, such as tensor decomposition. In this regard, Anaissi et al. [106] presented a tensor analysis-based damage detection method that allows for learning sensors interdependence. In this method, the acquired data is structured into a three-dimensional array with axes representing time, location, and frequency. A CANDECOMP/PARAFAC tensor decomposition is then used to obtain three matrices for features extraction. Finally, a one-class SVM (OCSVM) model is fitted using the features to detect novelties. The method is validated using two experimental case studies including a cable-stayed bridge instrumented with an array of 24 sensors. The method resulted in a damage detection accuracy of 92.5% compared to 61.1% achieved using the wavelet packet energy approach.

Döhler et al. [131] presented a subspace-based damage detection algorithm using residual vectors that are less susceptible to environmental changes combined with a generalized likelihood ratio test for anomaly detection. In a later study, Gres et al. [113] used the Mahalanobis distance (MD) of the empirical block-Hankel matrices constructed by the structure acceleration response as damage indicators. The method was validated using a numerical offshore mono bucket foundation and the S101 bridge, showing high sensitivity to low levels of damage but also suffering from a high false positive rate. The best results were obtained when fusing this approach with other subspace-based damage detection techniques with the use of Hotelling T^2^ control charts.

Sarmadi and Yuen [81] presented a one-class kernel null space algorithm based on the Foley–Sammon transform (FST), a linear subspace analysis method [160]. The damage index is the distance between the new observation null projection and the average of all training sample transformations in the kernel space. Based on extreme value theory (EVT), the generalized Pareto distribution (GPD) with a peak-over-threshold technique is used for threshold estimation. The kernel null FST-based novelty detector was tested on two bridges dataset, including the Z-24 bridge dataset, resulting in better damage detection performance compared to the OCSVM-MSD technique. It was also found that the inappropriate selection of the kernel and its parameters can impact performance by increasing the error rate.

### 3.2. Signal Processing Techniques

Signal processing techniques used in SHM involve time series analyses in the time domain, frequency domain, and time–frequency domain for extracting meaningful features from sensor measurements. Time series modeling (e.g., autoregressive modeling) is one of the most commonly used techniques for extracting time domain features. There are extensive studies that rely on time-series models to extract damage-sensitive features for SHM [31,161,162,163,164,165,166,167]. In time-series modeling, models’ parameters are often used directly as representative features, or further reduced using a dimensionality reduction method such as PCA. In other cases, the reconstruction error, also known as the residuals, is used to indicate damage. One of the main challenges with time-series modeling is the selection of model order, which impacts the damage detection accuracy. Researchers often propose a model order selection method that is based on regression accuracy or the model simplicity, such as the root mean square error or Akaike’s information criterion.

To study the effectiveness of different models used in damage detection, Shahidi et al. [130] compared the results of four different models: the single-variate regression, collinear regression, AR models, and autoregressive with exogenous input (ARX) models. For verification, a scaled steel frame test bed was used. The author showed that although all the methods were able to detect the damage, the ARX model had the best performance in localizing the damage.

Entezami and Shariatmadar [108] presented a damage detection, localization, and quantification method based on AR models’ parameters and residuals. An AR model was trained for each sensor record and the Ljung-Box Q-test was used as an iterative approach for model order selection. For the novelty test, they used the parametric assurance criterion (PAC) and residual reliability criterion (RRC) as damage indices, along with a 95% confidence interval. However, there are no instructions on how to combine both indices into a single novelty detection scheme. They validate their method using datasets of two experimental structures: the LANL three-story laboratory frame and the IASC-ASCE benchmark structure.

In a later study by Entezami et al. [102], they opted for extracting the features based solely on AR models residuals while using Partition-based Kullback–Leibler Divergence (PKLD) as a damage index. By relying on online learning, these adjustments made the damage detection system computationally more efficient and reliable under different environmental and operational conditions. Another, yet similar, feature extraction method was presented by Entezami et al. [88] based on autoregressive moving-average (ARMA) models’ coefficients and residuals. They also used a hybrid distance-based measure based on Euclidean-squared distance and PKLD with the nearest neighbor rule to indicate damage. More recently, Entezami et al. [89] trained an ARX model and used a nongraphical automatic model order termination method. The damage index in this framework is a hybrid distance-based measure combining PKLD with MSD using ARX model residuals. They validated their proposal on the Tianjin Yonghe Bridge, a cable-stayed bridge, and results showed that the method offers an improvement in efficiency over earlier methods. More researchers also favored time-series modeling for extracting the damage-sensitive features in the past decade [67,90,110,133].

Time–frequency domain representations are currently gaining more attention from SHM researchers as viable damage-sensitive features. Unlike frequency domain features, time–frequency features represent a signal record over both frequency and time. Short-Time Fourier Transform and WT are common techniques for extracting such features. Amezquita-Sanchez and Adeli [120] suggested using synchrosqueezed wavelet transform (SWT) combined with fractal modeling for damage detection, localization, and quantification. SWT was used to reduce signal noise, while fractal dimension (FD) is used to detect system changes using the median absolute deviation between the FD of training and new observations. The framework and three methods for estimating FD were tested on a laboratory-scaled 38-story building in Hong Kong [167], showing damage detection improvements by using SWT for denoising.

Ulriksen and Damkilde [127] presented a damage detection and localization method based on continuous wavelet transform (CWT) and a generalized discrete Teager–Kaiser operator. For detecting damage, they first applied PCA followed by MSD outlier analysis to detect abnormalities. Two case studies of a numerical beam model and an experimental study of a wind turbine blade showed that their method was able to localize introduced cracks. However, the method depends on a dense array of sensors which could make it impractical in some applications.

Xu et al. [98] introduced a two-level anomaly damage detection method based on WT, GPD, and moving fast Fourier transform (MFFT). WT was used to reduce the temperature effects on the raw measurements. GPD was used to estimate a more reliable threshold that corresponds to a 95% true detection rate within 100 years. They also implement a threshold updating strategy to include traffic volume increase and structure degradations. This anomaly test procedure is accompanied by anomaly trend detection that is based on MFFT. Their method was validated on a dataset from the Xihoumen Suspension Bridge with multiple numerically simulated anomalous and damage events. The method was successful in detecting most anomalies and a reduction in main cables stiffness, but failed to detect a reduction in the girder stiffness.

### 3.3. Signal Decomposition Techniques

Another method that deals with non-stationary signals is the empirical mode decomposition (EMD). Developed by Huang et al. [168], EMD is a data-driven method that iteratively decomposes the signal into simpler components, called the intrinsic mode functions (IMFs), which correspond to different oscillation modes. By examining the resulting IMFs, information regarding the signal, such as amplitudes and frequencies, can be obtained. Combined with Hilbert spectral analysis, more insights regarding the signal, and also a spectrogram, can be gained in a process known as the Hilbert–Huang transform (HHT). Several studies have conducted reviews and comparisons of different signal decomposition techniques in the context of fault and damage diagnosis [169,170,171].

Meredith et al. [140] examined the use of EMD for detecting and localizing damage in numerical beams based on the acceleration response of a moving load by detecting response discontinuity from the IMFs. They found that EMD can detect multiple cracks, but applying a moving average filter prior to EMD can make results easier to interpret. In another study, Kunwar et al. [135] explored bridge damage detection using a variety of output data from the HHT process also under a moving load. The experimental test structure was a small-scale single span-bridge instrumented with 10 wireless sensor nodes subject to three different levels of connection damage simulated by bolts removal. Results showed that the marginal Hilbert spectrum from a sensor in the proximity of damage can indicate a reduction in peak frequency compared to far away sensors. Additionally, changes in instantaneous phase values were more sensitive to simulated damage. Mohammadi Ghazi and Büyüköztürk [123] presented a damage diagnosis system based on HHT-based normalized cumulative energy distribution (NCED) and MSD-based hypothesis testing. Their framework combines four damage indicators estimated by comparing NCEDs of baseline and monitored structures with different methods, such as Kolmogorov–Smirnov distance. Although less efficient, the HHT-based approach provided better results when compared to a power spectral density-based method on a three-story laboratory steel frame. Striving for incorporating inter-channel information, Sony and Sadhu [70] propose using multivariate EMD for localizing structural damage. The absolute percentage change in energy from the baseline at each sensor is used as a localization indicator while the mean value among all sensors is used as an adaptive threshold. The method was capable of localizing damage in a numerical 10-DOF model and the Z-24 bridge despite limiting the number of sensors and observations.

EMD, however, suffers from the notorious mode mixing issue. Mode mixing occurs when the EMD process produces IMFs containing multiple frequencies, which should have been separated into individual IMFs. A number of modifications were proposed to solve this problem, resulting in ensemble EMD (EEMD), complete EEMD with adaptive noise (CEEMDAN), and variational mode decomposition (VMD). Xia et al. [118] used EEMD to separate temperature-induced strain from the raw strain measurements of a suspension bridge to identify damage. The temperature-induced strain was later used to form a matrix of Euclidean distance-based indices to facilitate damage diagnosis. However, the method does not involve a decision-making process. Soman [95] presented a semi-automated damage diagnosis framework for offshore wind turbine structures using EEMD and a sensitivity analysis-based thresholding method. The relative energy change in IMFs is used as a damage index, while the ratio of the individual sensors’ damage index to the mean of all sensors is used as a localization index. The method, however, is not entirely automated, as user input is needed at multiple stages in the framework, including the thresholding process, which may need access to historical damage data, making the method not fully unsupervised. Hybrid approaches involving EEMD for SHM were also developed, such as the EEMD-AR-ARX method proposed by Entezami and Shariatmadar [100] for damage-sensitive feature extraction accompanied by dynamic time warping for providing a dissimilarity measure.

Nevertheless, EEMD has some limitations, especially when the white noise amplitude is too low or excessive. Complementary EEMD (CEEMD), an extension of EEMD, alleviates this problem by using pairs of complementary white noise for signal decomposition. Tian et al. [103] combined the use of the Teager–Kaiser energy operator and CEEMD for bearing fault diagnosis. They argued that the proposed method is tailored to applications with weak vibration signals, as the Teager energy operator can enhance the signal’s strength before CEEMD can decompose the signal into a set of IMFs. An IMF is then manually selected and is further analyzed through envelope analysis to detect faults. It can be considered a feature extraction approach as no decision-making policy was proposed. Complete EEMD with adaptive noise (CEEMDAN) is also a variant of EEMD that relies on adaptive noise, which is updated based on the residue signal [172]. Mousavi et al. [91] explored combining the use of CEEMDAN and artificial neural networks for structural damage detection and localization. They trained a 20-layers deep network to predict four IMF features using IMFs as input. Three damage indices, estimated using the percentage error between healthy and new observations features output, are used to assess and locate damage.

An alternative method to EEMD that also does not suffer from the mode mixing problem, yet can be more efficient, is VMD [173]. Mousavi et al. [79] developed a VMD-based bridge damage diagnosis method under moving load by combining the instantaneous frequency and amplitude of the first IMF into a damage indicator. By testing their method on a numerical beam, they showed that VMD successfully localized damage when EMD could not. Similar conclusions were obtained by Sadeghi et al. [60], who compared the use of VMD to EMD for localizing shear connectors damage in composite beams based on shear slip data. The change in energy in the second mode center frequencies is used as a damage indicator. A laboratory-scale bridge fitted with slip sensors was used for evaluation, and the damage was simulated by unscrewing the shear connectors.

## 4. Unsupervised Learning SHM Based on Artificial Neural Networks

This section provides a summary of key studies in the past decade for using artificial neural networks (ANN) for SHM trained in absence of damage data. Basic ANNs have long been used for SHM. but there is currently a general trend toward leveraging deep learning techniques and architectures in building SHM. Like the feature extraction methods described in Section 3, ANNs are used to learn representations of the data and often reduce the dimensions. Deeper networks can even be used to extract representative features from raw vibration measurements without any preprocessing. A novelty detector is used for detecting system changes. which are usually simple tests depending on the network depth and the sensitivity of the learned features. In the field of SHM, some networks rely on supervised learning tactics for training. However, the labeling process is automatically performed through the available normal condition data only. Additionally, these methods incorporate representation learning and novelty detection strategies. We can therefore categorize them as unsupervised learning methods in the context of SHM, as per the categorization by Farrar and Worden [2]. Common learning goals include input reconstruction (e.g., autoencoders), forecasting (e.g., recurrent neural networks), and generative learning (e.g., generative adversarial networks). Examples of some of the commonly used ANN architectures are shown in Figure 13.

### 4.1. Classical Neural Networks

Classical ANNs are networks with relatively low parameter counts and no more than two hidden layers (Figure 13a). They are mostly used as dimensionality reduction techniques akin to those discussed in Section 3.1 (e.g., PCA). Avci and Abdeljaber [121] presented a structural health monitoring algorithm based on self-organizing maps (SOM). SOM are a type of ANN that can map high-dimensional input data onto a representative lower dimensional grid, often called topology maps, while preserving the topological structure of the data. The acceleration readings are used to construct the input matrix and the root mean squared error (RMSE) based on the topology maps of the baseline, and test data is used to identify damage. Their approach was tested on the phase II IASC-ASCE benchmark, showing a correlation between the damage index and the level of damage. However, no anomaly detection or thresholding methods were proposed. Gu et al. [114] proposed using an ANN for response reconstruction, which takes the temperature measurements as additional input in an attempt to reduce temperature variations effects. The proposed method uses the Euclidian distance between the predicted and target responses as an indicator of novelty. For verification, an experimental steel grid structure was tested under different temperature levels. The proposed method showed a good performance in differentiating between temperature changes and structural changes.

ANNs can also be used for forecasting to detect damage in structures. Neves et al. [116] proposed using an ANN trained to predict the upcoming acceleration values based on the structural acceleration response of passing trains. Then, the Gaussian process is used to provide a discordancy measure by categorizing the errors in the network predictions at each train speed. While the authors did not provide a clear-cut thresholding method, they suggested selecting the threshold based on the receiver operating characteristic (ROC) curves and false detection costs. ROC curves, however, are not easily obtained without access to damage observations. In a different damage detection approach, Movsessian et al. [80] proposed training an ANN which predicts the MD of the damage-sensitive features based on these features as input. Another damage indicator was proposed based on the network’s prediction error. They tested their method on a dataset captured from a wind turbine relying on the cross-covariance between the acceleration response as features.

Recently, Fernandez-Navamuel et al. [63] debated that traditional PCA is not ideal for data compression as it uses linear mapping, while kernel function selection for kernel PCA is challenging in an unsupervised setting. Therefore, they introduced an autoencoder-PCA hybrid that mimics the linear mapping of a PCA in addition to nonlinear residual connections between the low and high-dimensional feature layers. The reconstruction error of this hybrid network is used as a damage index and is tested against a threshold based on the 99th percentile of the baseline index values. The method was validated on two numerically simulated bridges calibrated with real-world measurements, and the results showed more accurate detections compared to linear PCA. This method is also capable of localizing damage if it was in proximity to one of the utilized sensors.

Different from autoencoders (AE), generalized autoencoders (GAE) make each input instance reconstruct a group of instances, not just itself. Li et al. [59] argued that GAE can better learn the basic structure of the original data while reducing noise effects compared to traditional AE. Therefore, they developed an SHM framework based on a modified GAE network which was trained to model power cepstral coefficients extracted from the structure response. The GAE was used to produce two damage indices in the form of normalized RMSE and the standard deviation ratio. For decision-making, they opted for MSD along with the 0.95 quantile of an F-distribution using training data. The method was validated using the Z-24 bridge dataset along with a numerically simulated dataset. Compared to traditional AE and PCA, the introduced GAE had a higher detection accuracy in the numerical case study.

### 4.2. Deep Dense Neural Networks

Multiple researchers opted for deeper architectures for their neural network models as advances are made in artificial intelligence and hardware technology (Figure 13b). With the added depth, the pre-network feature engineering steps can be further reduced, or entirely eliminated, by using direct acceleration measurements as input. These networks can also directly produce effective damage indices which facilitate the use of simpler novelty detection algorithms. Dense neural networks are used to refer to standard neural networks where all nodes in contiguous layers are connected forming a dense mesh of connections.

Ozdagli and Kooutsoukos [104] examined two unsupervised learning models; one relies on a dense AE to learn representation, and the other uses PCA. They used the Euclidean distance between the actual measurements and their model reconstruction as a damage indicator. Interestingly, temperature measurements can be added as an input parameter to both frameworks. They validated their methods via three case studies which showed that their method can detect and localize damage under temperature variability, especially when mode shapes are included as input parameters. Entezami et al. [90] presented a deep learning-based damage detection method with a focus on handling large quantities of high-dimensional data. The method combines ARMA coefficients and residuals as features, a deep AE as a dimensionality reduction mechanism, and MD as a novelty detector into a single framework. A final prediction error function is used to optimize the number of nodes in the deep autoencoder layers. Generalized extreme value (GEV) distribution with block maxima technique is used for thresholding. The method was able to accurately detect damage cases in the Tianjin Yonghe cable-stayed bridge dataset. In another study that also advocates for automatic feature extraction, Jiang et al. [75] introduced two deep AE architectures for learning damage-sensitive features from raw acceleration measurements. The first provides features from the bottleneck layer while the other uses the reconstruction error as features. Multiple AE networks are trained in parallel, one for each sensor. The learned features are then tested for detecting structural damage against a predefined threshold, which they suggest setting according to the structure’s importance. Localization is also possible based on which sensor the anomalous features were extracted from. These methods are validated using both the LANL three-story frame structure and QUGS.

Silva et al. [83] introduced a damage-sensitive feature extraction method using stacked AEs. They used natural frequencies as input to their network and used the output of the bottleneck layer as a more compressed damage-sensitive feature vector, often two features only. While their proposal did not include a novelty detection method, they used Gaussian mixture models (GMM) based on expectation-maximization as an example. A comparison with other representation learning techniques including PCA, AANN, and kernel PCA was performed using the Z-24 bridge dataset. Their model provided the best damage detection results, which were slightly better than kernel PCA while requiring fewer parameters for the features vector. However, the explainability and the physical interpretation of the introduced method are low compared to raw modal parameters and other classical methods.

Exploring deep-learning solutions for unsupervised learning SHM, Wang and Cha [85] introduced a deep dense AE network to automatically extract features from raw acceleration. Three metrics are used to produce the features: MSE, original-to-reconstructed-signal ratio (ORSR), and Arias intensity. Two or more of these metrics are fed into an OCSVM model for novelty detection. However, it requires predefined hyperparameters which control the shape of the decision boundary, affecting the decision-making process. The method is validated using two case studies, including a laboratory-scaled steel bridge where the method detected a 10% stiffness reduction. It is, however, not suitable for damage localization and quantification. Using the same metrics, Giglioni et al. [64] introduced an ensemble-based damage detection and localization method for large-scale structures based using AE models. An AE network is trained for each sensor based on raw measurements and MSE and ORSR metrics are obtained from each model, forming a binary decision matrix. A value of 1 is given at a sensor and index when it passes a threshold based on the 90th percentile; otherwise, it remains zero. A summation-based ensemble inference method can then be used to assess global damage in addition to localization capabilities. The method was validated using the Z-24 bridge dataset showing promising damage detection performance with fair localization ability.

### 4.3. Convolutional Neural Networks

A convolutional neural network (CNN) is a type of deep neural network that relies on convolution operation using learned filters that allow for weight (Figure 13c). This makes CNNs demand fewer parameters than deep dense neural networks, making them ideal for deeper networks. CNNs are generally best suited to grid-like structured data with local spatial correlation, such as images. Therefore, to use vibration data for CNNs, researchers often propose a data organization method to make full use of CNN’s capabilities [58].

Focusing on data compression for SHM, Ni et al. [93] introduced two deep convolutional autoencoder models for detecting measurement anomalies and compressing the recorded data. Both networks rely on 1D-CNN autoencoder architectures. The first is used for anomaly detection and is trained in a supervised fashion, while the second is for data compression and reconstruction. The method was tested using a dataset from a suspension bridge showing good compression and reconstruction performance when the compression rate is 0.1. Shi et al. [68] developed two forecasting dense-based and CNN-based neural networks for real-time SHM. A noteworthy feature in their framework is the use of model pruning to make their networks more efficient by eliminating some of the redundant connections between neurons with insignificant loss to model accuracy (Figure 14). The prediction errors of their models are used as features and deep support vector domain description (SVDD) generates a decision boundary and is used as a novelty detector. Their method is tested using two case studies of frame structures, including an experimental study, showing its real-time damage detection performance. Rastin et al. [87] presented a damage detection framework based on a convolutional AE (CAE) trained on a matrix formed of stacked acceleration measurements. They use the Euclidean distance between the latent vectors of baseline and unknown structural states as damage indicators after normalized them to unit vectors. Novelty detection is performed by setting a threshold, typically 1.6 or 1.4 standard deviations away from the mean. The Tianjin Yonghe Bridge and two numerical case studies are used for evaluation.

Variational autoencoder (VAE) is a type of variational inference-based generative model that treats the latent features as random variables with a prior distribution. The network learns the latent distribution during training which can be later used to generate new samples. Ma et al. [91] proposed a bridge damage localization approach based on a one-dimensional convolutional variational autoencoder (CVAE) as a dimensionality reduction method. The model input and output are the acceleration response to a moving load, and the Euclidean distance between the latent features at different time steps is used as a localization index. However, since this approach only localizes damage, no thresholding strategy was proposed. Another one-dimensional CVAE architecture was proposed by Yuan et al. [86] to identify light rail squat damage, and they combined it with either an MSD-based elliptic envelope or OC-SVM as an anomaly detector. When tested on a laboratory full-scale track platform, they concluded that the elliptic envelope was the better choice, as it makes full use of the Gaussianity of the latent variables. Kim et al. [65] presented a CVAE-based damage localization system that encodes the structures’ flexibility matrices obtained by operational modal analysis. A flexibility disassembly method is then used to localize damage by comparing the input and output of the CVAE model. Estimating the flexibility matrix, however, can be challenging for complex structures. Zhang et al. [73] developed an unsupervised tunnel damage detection method using wavelet packet energy of trains’ dynamic response data, which are fed into a CVAE. The RMSE is used as a damage index, while the relative entropy of wavelet packet energy is used to localize damage. The method was tested on a laboratory-scale tunnel, resulting in 91.5% accuracy. It was not, however, tested using data with a low signal-to-noise ratio.

### 4.4. Other Deep Learning Architectures

There are different other types of neural network models used for unsupervised learning SHM beside those discussed earlier. A restricted Boltzmann machine (RBM) is a stochastic neural network that can be used for dimensionality reduction. Rafiei and Adeli [109] introduced a damage detection and localization method tailored to high-rise buildings based on a deep RBM architecture. The hidden nodes of the RBM are used to estimate a structural health index. The building is split into multiple parts, which are used for training parallel models to localize damage. Their method is validated using the dataset of the laboratory-scaled 38-story building in Hong Kong [167]. Based on the results, they provide recommendations for health index ranges corresponding to different damage levels. However, it does not seem that the method incorporates a thresholding scheme based solely on undamaged data.

Graph convolutional network (GCN) is a generalized version of CNNs where node connectivity is predefined (or learned) via a global adjacency matrix instead of the standard local connections. It is also a part of the graph neural networks family. Li et al. [76] argued that it is not ideal to represent vibration data in image structure form and that the sensor’s spatial correlation is not easily learned by CNNs. Instead, they proposed using a spatiotemporal GCN for sensor fault detection to learn both the spatial and temporal dependencies of the sensor measurements. The network adopts GCNs with trainable adjacency matrices in addition to temporal 1D-CNNs. The framework was tested on a dataset of cable forces from a cable-stayed bridge, and it used the learned adjacency matrices and the model residuals to detect faulty sensors in a novelty detection scheme.

A long short-term memory (LSTM) network is a type of recurrent neural network that is commonly used to learn complex temporal patterns from time-varying data [174,175]. Son et al. [84] proposed a two-stage anomaly detection framework, relying on an encoder-decoder LSTM network, for identifying abnormalities in the collected SHM data. Since their focus was on monitoring cable-stayed bridges, the input to their two-layered LSTM network is the raw cable tension time series, and the reconstruction error is used for estimating an anomaly score. While their method provided an ROC of 0.99, it did not identify the anomaly source, which could be structural damage, sensing malfunctions, or environmental effects. Eltouny and Liang [58] presented a spatiotemporal composite autoencoder network for detecting and localizing damage in systems with a large sensor array (Figure 15). The network is a CNN-LSTM hybrid network with a dual output providing both a signal reconstruction and a forecast, making it suited for learning spatial and temporal dependencies in the data. Raw accelerations are organized into a grid-like structure incorporating the sensor’s location and the time-domain, which are then used as an input to the network. Damage indices are obtained from the latent features, and the output residuals and novelties are identified using an EVT-based threshold. The framework achieved accuracies of 93.1% and 85.2% for damage detection and localization, respectively, when tested on a numerical multi-bay, multi-story structure. The case study also showed that results were not significantly impacted by the reduction in available sensors.

Generative Adversarial Networks (GAN) are generative models composed of a generator and a discriminator neural network that compete against one another during training (Figure 13d). The generator learns to generate synthetic samples that are realistic enough that the discriminator fails to identify them as synthetic. The weights of the two models are updated iteratively based on their performance. Mao et al. [78] proposed an anomaly detection algorithm for structural health monitoring, which combines a convolutional GAN and a CAE into a single model. The vibration data are converted into Gramian Angular Field images, to be better suited for the CNN layers, and are used as input to the hybrid network. After training the GAN model, the trained generator is set as a decoder for the CAE. For evaluation, the latent features and cumulative sum control charts are used to detect anomalous data in a cable-stayed bridge, achieving more than 94% accuracies for all channels. With a focus on the sensitivity to sensors’ configurations, Soleimani-Babakamali et al. [69] introduced three GAN models to be used in a damage-detection framework (Figure 13d). The three models share the same dense generator architecture but have different discriminators, including dense-based, CNN-based, and LSTM-based models. All GAN networks use normalized FFT amplitudes as input and provide a discriminator score which is used as a damage index. One of the interesting aspects of this method is the threshold tuning and adaptive thresholding, making it capable of detecting recurrent novelties. Their models were tested using both QUGS and the IASC-ASCE benchmark structure, and it was concluded that the LSTM-based GAN provided the best damage detection results. In a later study, Soleimani-Babakamali et al. [61] investigated the effects of dimensionality reduction on the damage detection results using the models proposed in [69] after applying techniques such as PCA, kernel PCA, and AE. It was found that reducing the dimensions of the input vector had a negative impact on the detection accuracy, but regularization of nonlinear methods can reduce this effect.

## 5. Novelty Detection Techniques

The majority of vibration-based unsupervised-learning SHM methods rely on novelty (or anomaly) detectors for detecting system changes based on damage-sensitive features (Figure 16). Sometimes more than one novelty detector is used in a stacked fashion to further compress the input features, or in an ensemble learning framework. Most novelty detection techniques theorize that the normal condition forms a single class (or a cluster), and points outside the class’s boundary are flagged as novelties, anomalies, or outliers. This makes this type of novelty detection a one-class classification problem, and examples of this approach include outlier analysis and OCSVM. If the data forms multiple groups, due to multiple normal or damaged conditions, then cluster analysis is preferred. In this section, we provide a summary of key studies in the last decade with a focus on the novelty detection aspect of the unsupervised-learning SHM framework.

### 5.1. One-Class Novelty Detection

One-class outlier analysis is by far the most common novelty detection technique used in the unsupervised learning SHM literature. Even when performing cluster analysis or processing the features using complex deep learning techniques, a simple significance test is often performed to the estimated damage index for a deterministic decision. In one-class novelty detection, all obtained damage indices from the training data are used to fit a distribution, most commonly the Gaussian distribution, representing the normal class. Damage cases are detected when the index exceeds a certain threshold, which is set using various statistical techniques.

For univariate outlier analysis, a statistical significance test (e.g., z-test or *t*-test) is commonly used. A threshold can be set using confidence intervals [83,108,130,136], significance [62,73,88,102,120], percentiles [58,63,64,66,74], or other data statistics. For multidimensional features, MD, or MSD, is often used [59,67,72,82,89,90,99,116,132]. There are different techniques for selecting a threshold for MSD-based outlier detection. One popular method is based on Monte Carlo simulation, which is described by Worden et al. [29,72,99,130]. One advantage of this technique is that it takes into account the size of the data and the feature dimension besides the chosen percentile. Another way is to assume that the data follows a Chi-square distribution with degrees of freedom corresponding to the feature dimension [67]. Related to the MSD-based outlier analysis, Hotelling T^2^ control charts are also used for detecting novelties [96,105]. For outlier analysis in both its univariate and multivariate modes, some researchers rely on EVT for selecting the threshold [58,62,74,81,82,90].

Nigro et al. [133] provided a comparison of different damage detection statistics with a focus on damage localization. Three statistics were inspected: the cumulative sum indicator, the exponentially weighted moving average, and a modified MSE metric which is normalized according to data variance. For each metric, they proposed an outlier detection based on a confidence interval produced via a bootstrapping process. They also suggested combining all metrics into a multivariate feature using MD with the Fisher Criterion to provide the localization threshold.

Besides outlier analysis, other one-class novelty detection methods were adopted for unsupervised learning damage detection frameworks. OCSVM is a novelty detection method used in SHM frameworks that attempts to learn a decision boundary around the training data [85,106]. It first maps the data into a higher dimension using a kernel function, then finds the maximum marginal hyperplane which separates the data from the origin [176]. A further one-class classifier that is being used for damage detection is SVDD, which tries to find the hypersphere with the minimum radius that encloses the training data [68,177].

### 5.2. Cluster Analysis

Clustering is a statistical modeling technique that aims to sort observations with similar features in groups or clusters. Commonly used clustering techniques are classified into partition-based (e.g., K-means), hierarchy-based, distribution-based (e.g., Gaussian mixture models), fuzzy theory-based (e.g., fuzzy c-means), and density-based (e.g., density peaks) [178]. In SHM, clustering is typically used as a decision-making algorithm in which models extracted damage-sensitive features to detect abnormalities. Examples of using clustering for novelty detection include testing if a new observation does not belong to any of the modeled clusters and if a group of new observations can instead form their own cluster. Nevertheless, it is sometimes used for feature extraction and reduction followed by a one-class novelty detector.

K-means clustering is a partition-based clustering technique and one of the simplest and most widely used in SHM. In this method, each data point is assigned to one of the *k* number of clusters with the closest centroid. Its simplicity and efficiency are what attract many researchers to apply it in unsupervised learning-based SHM frameworks. However, it may need advanced feature engineering in the earlier stages of the framework before clustering. Diez et al. [122] suggested performing the FFT algorithm on the collected dynamic response data to improve efficiency. Moreover, they removed the outliers first, using the k-nearest neighbor algorithm. Then by applying the k-means algorithm to the extracted features, the abnormal conditions can be detected. The proposed method was verified by the Sydney Harbour Bridge benchmark. Santos et al. [124] also used k-means clustering to detect stiffness reductions from a cable-stayed bridge. They used the global silhouette index for cluster validity and the Gowda–Diday dissimilarity measure as a damage index. More recently, Meixedo et al. [67] introduced a clustering-based SHM method that relies on the transient response from train passing to detect bridge damage under environmental conditions. The damage-sensitive features are the parameters of ARX models reduced by PCA and further processed by MD. The resulting features are fitted to clusters by k-means and the average dissimilarity between clusters is used as a damage index.

There are many other variants of the k-means algorithm which attempt to address the method’s shortcomings. The k-means—algorithm is a clustering algorithm modified from the k-mean clustering algorithm [179]. Where the conventional k-means is sensitive to outliers, the k-means—method overcomes this limitation by removing the clusters that have only one member. Alamdari et al. [112] proposed using a modified k-means—algorithm for detecting damage. The verification was performed using the Sydney Harbour Bridge benchmark, and it was found that the proposed method was able to detect the abnormal responses in the damaged arches. Another method that is related to k-means clustering is the k-medoids algorithm. Unlike k-means, k-medoids uses an actual datapoint as a cluster center and minimizes the dissimilarities between points within a cluster, making it more robust to outliers. De Almeida Cardoso et al. [100] used the interquartile ranges and the medians in both time and frequency domains (obtained using FFT) as damage-sensitive features. They then used the k-medoids clustering method to detect novelties based on the distances between all possible medoids. A threshold is set based on the 99.9th percentile of a t-student distribution measured from the novelty index median. Both the IASC-ASCE benchmark structure and the PI-57 bridge were used as experimental case studies. The results showed that, with tuned hyperparameters, the method is successful in detecting low levels of damage under environmental and operational conditions. However, these hyperparameters may be difficult to tune in an unsupervised learning setting.

FCM, also known as soft clustering, is a fuzzy theory-based clustering algorithm that shares a lot of similarities with k-means. Instead of assigning each data point to a unique cluster, FCM provides a grade of membership ranging from 0 to 1 to all clusters. In this case, data points can potentially belong to multiple clusters to a certain degree. FCM was first proposed by Dunn [180], improved by Bezdek [181], and was applied to SHM applications in the past few decades [36]. In 2013, Yu et al. [138] introduced a damage detection approach based on reduced frequency response function and fuzzy c-means clustering. They also suggested using either PCA or kernel PCA for dimensionality reduction. Their method was validated on a steel truss bridge subject to different damage scenarios simulated by loosened bolts. Alves et al. [128] suggested a monitoring method using symbolic signals and clustering techniques. First, he manipulated the raw dynamic response data using symbolic analysis, and then applied three clustering techniques: dynamic clouds, FCM, and hierarchical clustering. The proposed methodology was verified using data collected from the PI-57 bridge, showing better results achieved by FCM compared to the other two clustering methods.

Hierarchical clustering is a greedy clustering algorithm that establishes a hierarchy of clusters for data points based on their inter-similarity. This can be performed either in a bottom-up manner (agglomerative) where each point starts as its own cluster and merging is performed at each step, or in a top-down manner (divisive). In 2017, Zhou et al. [119] proposed a hierarchical clustering model to detect damage in structures. The proposed model takes transmissibility as an input feature. Two similarity measures were adopted for damage indication: cosine similarity and distance similarity. In a later study, Tran and Ozer [97] introduced a bridge health monitoring framework using modal parameters along with hierarchical clustering. They used a univariate anomaly detection method based on the gaussian distribution as a discriminant test and validated their method on a laboratory bridge experiment and a steel pedestrian bridge. However, they argued that it is difficult to properly select the clustering threshold as cross-validation is not easily performed with the absence of damaging data.

GMM is a probabilistic model that assumes all the data points are generated from a mixture of a finite number of Gaussian distributions with unknown parameters. Figueiredo and Cross [134] compared MSD-, PCA-, auto-associative neural networks-, and GMM-based novelty detection methods for bridge damage detection under the influence of operational and environmental variabilities. When evaluating these methods using the Z-24 bridge, they found that the MSD of GMM parameters as a damage indicator provides the least errors among the other three methods. They also conclude that linear methods, such as PCA and plain MSD, struggle to remove the nonlinear patterns caused by the operational and environmental effects, leading to a high number of false positives. Santos et al. [117] used GMM for clustering with the expectation-maximization (EM) algorithm to detect the anomalies. EM is dependent on the initial guess of the parameters. Thus, the authors used the genetic algorithm along with EM to improve the overall performance of the system. Using the Z-24 bridge benchmark as a validation, it was shown that introducing the genetic algorithm improved the stability of the EM method, especially in minimizing type 2 errors. Addressing tie-rods evolutive damage, such as corrosion, Lucà et al. [66] proposed a tie-rod damage detection method by fitting a GMM using eigenfrequencies. The existence of damage can be detected based on the likelihood values of two GMM hypotheses, which are single versus double Gaussian densities. An experimental setup of tie-rods was used to validate this method and it was concluded that the GMM-based method outperformed the MSD-based one in detecting evolutive deteriorative phenomena, but MSD could be more suitable to sudden damage scenarios.

There are many other variations of clustering algorithms. For example, Silva et al. [125] presented a genetic algorithm-based clustering method for unsupervised learning bridge health monitoring. Selecting the number of clusters is often challenging, and thus they rely on a concentric hypersphere algorithm to optimize the number of clusters. The minimum Euclidean distance between new observations and cluster centroids is used as a damage index. The method outperformed GMM and MSD-based outlier analysis for damage detection when applied to two case studies: The Z-24 bridge, and the Tamar bridge. Spectral clustering (SC) is a clustering technique based on graph theory that utilizes the eigenvalues of the similarity matrix. Kernel spectral clustering (KSC) is the kernel-based variant of spectral clustering, making it a useful algorithm for clustering data that is not linearly separable. It is also useful for handling large data sets, as the computational cost of SC can be reduced by using an appropriate kernel function. Langone et al. [115] proposed a damage detection method based on an adaptive KSC algorithm and validated it using the Z-24 benchmark dataset. The calibration of the model is performed during the undamaged case, and then it can be applied to detect the anomaly. Density-based clustering algorithms generate clusters that are characterized by centers of high observation density in the feature space. Cha and Wang [107] modified the density peaks-based fast clustering algorithm to train under an unsupervised learning setting for damage detection and localization. They used features based on continuous wavelet transform (CVT) and the crest factor. In the testing phase, observations with a local density below a predefined cut-off are considered novel. Using a laboratory-scale steel structure, the method was able to outperform OCSVM in damage localization. However, it was found to be computationally expensive, and recommendations were given to increase its efficiency.

### 5.3. Bayesian Methods

Bayesian analysis relies on the Bayes theorem to update probabilities based on prior information. Bayesian methods interpret probability as a degree of belief and can often be used to incorporate uncertainty in parameter estimation. Sankararaman and Mahadevan [137] focused on quantifying the uncertainty for the detection, localization, and quantification of damage using Bayesian approaches. For damage detection, they used Bayesian hypothesis testing of the model residuals to estimate the Bayes factor. Then the limits set by Harold Jeffreys [182] were used to assess damage based on the Bayes factor. The Bayes factor can later be used to estimate the probability for each of the two scenarios which can be treated as an uncertainty measure. They also quantified the uncertainty for both damage localization and quantification using the concepts of likelihood and Bayesian inference. They also presented a strategy for updating the uncertainty with the acquisition of new measurements. Wang et al. [71] also used Jeffreys–Bayes factor hypothesis testing to detect the structural damage in the Tianjin Yonghe cable-stayed bridge. They proposed a damage index obtained from the Natural Excitation Technique, which has both real and imaginary parts. A sparse Bayesian learning regression model was then trained to predict the imaginary part given the real part as input. The relative change between the two parts was used to assess the structural health condition. They also proposed using Bayes factor as an indicator of damage severity.

Yan et al. [72] introduced an unsupervised method for damage detection which also attempts to accommodate uncertainties, such as data randomness, measurement error, and environmental variability. Damage is detected by estimating the symmetric KL divergence between the transmissibility function (TF) of a baseline condition and the TF of an unknown condition. They use a statistical threshold estimation process involving Bayesian inference and Monte Carlo discordancy testing [29] to account for the measurement uncertainty. This framework is validated through four case studies, one of which uses experimental data from the S101 bridge. Results show satisfactory performance in detecting global damage and quantifying its severity. However, damage localization was not possible, as the locations of anomalies-producing sensors do not necessarily correspond to the damage location.

Bayesian approaches can be used for parameter estimation. For example, Figueiredo et al. [132] introduced a Bayesian approach based on Markov-Chain Monte Carlo for GMM clustering instead of the conventional EM method. The novelty test was carried out using MSD-based outlier detection. While results on the Z-24 bridge dataset showed comparable performance to the EM-based clustering method, the Bayesian approach offered some insights for the model which, for example, aided in the selection of the number of components. In another study, and aiming to avoid predefined data distributions, Eltouny and Liang [74] used Bayesian optimization to build a probabilistic model based on the kernel-density maximum-entropy (KDME) method for localizing damage. Bayesian optimization is a global optimization technique often used to tune machine learning models’ hyperparameters without assuming the form of the objective function. To train a multivariate KDME, they relied on independent component analysis as a preliminary step. Joint probabilities of new observations are used as a damage index, and a threshold is set based on EVT. The method was validated using three case studies of a three-story concrete building, a high-rise structure, and an experimental masonry frame, achieving an average accuracy of 92.6%. It was also found that Bayesian optimization significantly accelerated the tuning of the KDME model compared to the genetic algorithm.

### 5.4. Other Methods

In addition to the aforementioned machine learning techniques for detecting system changes, other methods are discussed here including robust regression, ensemble learning, and empirical machine learning. Robust regression is a method to estimate mathematical model parameters while minimizing the effect of outliers. Dervilis et al. [129] applied robust regression using the least trimmed squares (LTS) and the minimum covariance determinant (MCD) algorithms on the Z-24 and Tamar bridges datasets to detect structural damage. The author suggested that future research can study moving from linear LTS to non-linear robust regression.

Ensemble learning is a machine learning technique that combines a set of models in a way that promotes diversity to improve and stabilize predictions. There are many ways to create an ensemble, such as bootstrap aggregating and stacking. Inference can also be performed in different ways, such as voting or averaging. Multiple researchers used ensemble learning for improving the damage detection performance of novelty detectors. For example, Bull et al. [99] presented an ensemble of MSD-based novelty detection models for SHM to reduce the masking effects produced by inclusive outliers. The ensemble models were generated using the bootstrap sampling technique and the models’ averages were used as the ensemble output. For thresholding, they used the Monte Carlo simulation thresholding method [29]. When compared to MCD [128] using the Z-24 bridge dataset, the outlier ensembles provide comparable results with a significant reduction in computational cost. Both the presented method and the MCD benchmark, however, produced false positives for the Z-24 bridge during the cold weather monitoring period. The method was also tested on an aircraft wing for damage localization, achieving 95.85% detection accuracy in an unsupervised learning setting.

Another ensemble learning-based unsupervised learning damage detection technique was presented by Sarmadi et al. [82]. They aimed to benefit from the computational efficiency of ensemble learning while also mitigating environmental variability effects on the SHM system. Their sequential learning framework included three different MD variants. A set of nearest neighbors of the features are obtained at each level using the distance participation factor. The local MSD values at the final level are used as damage indices, and a threshold is set based on EVT. The method was validated using two experimental case studies, including the Z-24 bridge, and it successfully detected damage under strong environmental variations. It also produced a lower error rate when compared to a selection of traditional techniques, such as PCA, k-means clustering, and MSD.

Multiple-model (MM) learning is a method closely related to ensemble learning where more than one statistical model is used to analyze or make predictions about a dataset. Vamvoudakis-Stefanou et al. [110] compared two AR methods based on MM models with two other conventional autoregressive models. They used MM to represent the undamaged dynamics of a structure with a set of conventional models using estimated parameter vectors and Gaussian probability density functions. For assessment purposes, ROC curves were used to represent the accuracy of each model. The study included a population of 31 composite beams subjected to impact damage at two different energy levels. The results showed the MM-based models achieved significantly improved results, especially for low-energy-level damage, where all damage is correctly detected at an error rate of 5%.

Some methods opt for the more flexible non-parametric methods which do not impose prior assumptions on the data. Empirical machine learning only relies on the observations and the relative distance between them for building models [183]. Using this concept, Entezami et al. [62] introduced a damage index obtained by multiplying the empirical local density by the minimum distance of each sample to all other samples. This non-parametric novelty detection approach was inspired by the density peak clustering method [107]. When applied to both the Z-24 and Tianjin Yonghe bridges, the method outperformed a selection of other non-parametric novelty detection techniques in damage detection and computational efficiency.

## 6. Challenges and Future Trends

While unsupervised learning offers a more practical approach for applying vibration-based SHM compared to its supervised counterpart, some limitations and challenges are delaying widespread industrial use. Most of these difficulties stem from the concept of unsupervised learning SHM, that is, the absence of classes of both damaged and undamaged conditions. The section summarizes the current challenges in unsupervised learning vibration-based SHM applications as well as future research observed from the reviewed literature.

### 6.1. Parameters Selection

With the absence of damaged and some undamaged classes, performing cross-validation for parameter tuning is challenging. This is especially true for thresholding, as the selection of a significance level for outlier analysis, the parameter “*ν*” in OCSVM, or the number of clusters is nontrivial in an unsupervised learning setting [97,99]. Instead, they are often selected based on engineering judgment, which would not necessarily provide the optimal SHM model. In novelty detection, it is mainly a problem of balancing the false positive and false negative rates according to the design objective. Before testing, a boundary can be established to fit the available normal data with the assumption that the training data is a representative sample of the normal condition. Nevertheless, the size of the data may not be large enough to establish the boundary, especially if there are no available observations for other normal classes or if there is an overlap with the unknown damaged class. The damage detection performance of some models can be less sensitive to parameter selection, yet post-test sensitivity analysis may show superior attainable models [74].

Multiple attempts have been made to provide a more robust threshold via Monte Carlo sampling techniques [72,99,129,133], while others implemented an EVT-based threshold selection procedure [81]. Stochastic and EVT-based thresholding methods, however, may also need selecting parameters, such as the sample size or the block maxima window size. Others have provided threshold selection guidance based on case study results of post-test ROC curves [116]. On the other hand, most proposed SHM methods are structure-specific and suffer from a generalization problem. Adaptive thresholding methods have also been proposed such that the novelty detector can identify future damage scenarios when the system was already subject to change [67,69]. The threshold selection strategy is expected to remain an open area of research in the future.

### 6.2. Environmental and Operational Variability

Environmental and operational variations pose a major challenge to the development of SHM methods in general, and unsupervised learning-based methods in particular [184]. Novelty detection attempts to detect deviations from normality in the system based on damage-sensitive features. These deviations could be attributed to structural damage or other system changes such as variations in the temperature or operational conditions. For example, model parameters, which are commonly used as damage-sensitive features, can drastically change due to temperature variations [185]. In the past decade, many researchers focused on developing damage-sensitive features and novelty detection techniques that are less sensitive to environmental variability [66,67,81,82,83,96,105,125]. Others considered a probabilistic novelty detection method that would incorporate uncertainties including environmental variations and measurement noise [72,132]. In addition, expanding the training dataset such that it includes a wider range of environmental conditions is important to reduce these effects and the resulting false positives [74,186]. Additionally, including measurements other than vibration to the SHM model input, such as temperature, wind speed, and loads, can add valuable information to the model and help reduce the uncertainty associated with damage identification [104,114].

Recent literature suggests that mitigating the effects of environmental and operational variability remains a key topic in SHM systems development. We expect that future unsupervised learning SHM research attempts to tackle this challenge by (1) utilizing the advances in sensor technology and the internet of things to collect long-term monitoring data to decrease the data uncertainty; (2) relying more on unsupervised learning methods that can learn highly nonlinear and nonstationary patterns, especially deep learning methods (see Section 6.5); (3) incorporating environment monitoring systems (e.g., temperature, humidity, and wind speed sensors) and performing a fusion of data from different sources, which can add extra layers of information.

### 6.3. Benchmarking Standards

While there are various types of unsupervised learning SHM frameworks, it appears that there is no standard practice for establishing a direct comparison between them. Benchmarking practices often exist in domains with other machine learning applications, such as using ImageNet [187] for computer vision models or Human3.6m [188] for human sensing models. While some benchmarks are available in the literature (as described in Section 2), comparisons with previously introduced methods are lacking because authors often test their methods using different metrics and partitions of the dataset [184]. Taking the Z-24 Bridge dataset as an example, some authors use the undamaged state period from 11 November 1997 to 4 August 1998 for training, while the damaged state period from 5 August to 10 September 1998 is used for testing [83]. Others use 75% of the undamaged state period for training while adding the rest to the testing set [81]. Some alternative methods are including the first month, the first three months, or the last undamaged state month of the record for training, in addition to different sets of testing [59,115]. In addition, making the model source code available for the community, which would facilitate the comparisons or the building of a Model Zoo, is still not a common practice in SHM research. Standardized benchmarks, in general, can provide a quick overview of the state-of-the-art and may accelerate the development of SHM machine-learning models. Therefore, it is encouraged that this practice is adopted in the SHM future research.

### 6.4. Datasets Availability

As per the reviewed literature, there are multiple available datasets that can be used for the validation of the proposed methods. However, most of these datasets fail to represent real cases of damaged structures. Figure 17 shows the percentage of datasets used from field studies with real observed damage, laboratory tests, and numerical simulations, based on the reviewed literature in Table 1. Many of these methods are based on numerical simulations and laboratory experiments, with the latter being an improvement over the former. Datasets based on real structures which suffered damage, such as the commonly used Z-24 bridge dataset, would often record damage cases without consideration of real-life operational conditions (e.g., passing traffic). The collection of datasets that include structural states under realistic conditions and uncertainties while being large enough to train deep learning models remains desired for developing advanced unsupervised learning SHM methods.

### 6.5. Deep Learning

Deep learning-based SHM methods have gained considerable attention in the last few years, as demonstrated in Figure 2. Nevertheless, they require a significant amount of training data, and the limited availability of experimental data can be an inhibiting factor for the development of more complex architectures. This is especially the case for supervised learning methods, but obtaining large quantities of training data for unsupervised learning SHM frameworks can be easier. It is expected that more deep learning-based unsupervised learning SHM methods will emerge in future studies with the advent of big data benchmarks.

### 6.6. Model Generalization

While unsupervised learning offers a more practical solution to SHM compared to supervised learning, it still suffers from limitations that are slowing the transition to industrial practice. Compared to machine condition monitoring, which benefits from reliable statistics obtained from similar applications, unsupervised learning SHM methods are mostly structure specific, as civil structures often have unique characteristics [51]. The generalization of vibration-based SHM methods is therefore needed. In recent years, some researchers have attempted to address this problem by proposing methodologies based on the concept of transfer learning (or domain adaptation) [189,190,191,192] and self-supervised learning [193]. It is expected that interest in this topic will keep increasing in future research, especially with the rapid advancements in deep learning research.

## 7. Conclusions

SHM is an important asset for autonomous, real-time structural condition assessment. Unsupervised learning could be the key to closing the gap between academia and industry for vibration-based SHM. This study provides a detailed review of the state-of-the-art unsupervised-learning SHM applications in the past decade. These methods involve different types of unsupervised learning techniques, including conventional feature extraction techniques (e.g., PCA, AR models), deep learning methods (e.g., AE, GAN), novelty detection, and cluster analysis. Additionally, a selection of common benchmarks used in unsupervised learning SHM were described. Challenges, such as thresholding, environmental variability, and model generalization, were discussed based on the reviewed literature. In summary, it is expected that more unsupervised learning SHM techniques will be developed in the upcoming years that will attempt to address the described challenges with practicality as a primary objective.

## Figures and Tables

**Figure 1 sensors-23-03290-f001:**
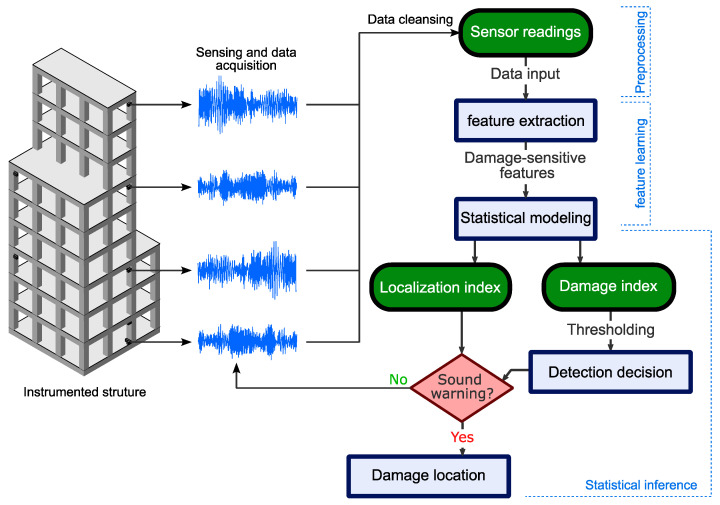
An example flowchart of a data-driven structural damage detection system.

**Figure 2 sensors-23-03290-f002:**
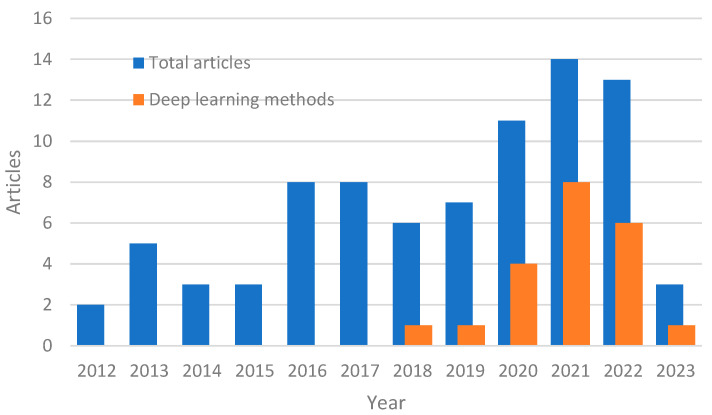
The number of reviewed unsupervised learning-based SHM studies using deep learning and the total reviewed articles over the years since 2012.

**Figure 3 sensors-23-03290-f003:**
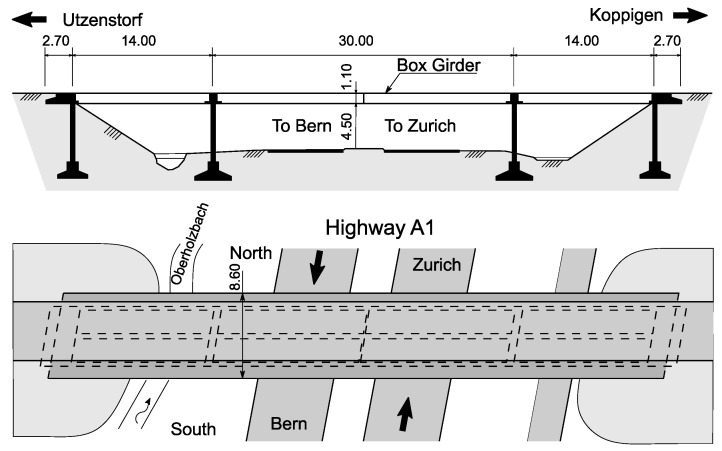
Bridge Z-24 schematic drawing (units are in meters; adapted from [142]).

**Figure 4 sensors-23-03290-f004:**
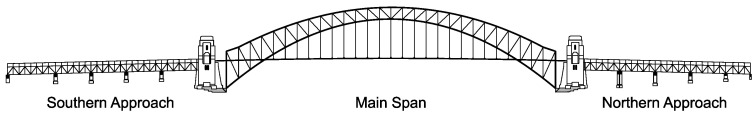
SHB schematic drawing (adapted from [112]).

**Figure 5 sensors-23-03290-f005:**
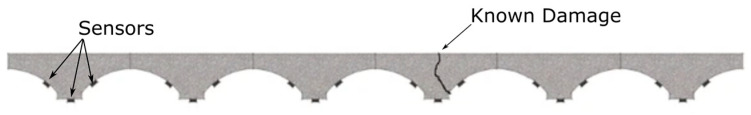
Sample of sensors set up on SHB [112].

**Figure 6 sensors-23-03290-f006:**
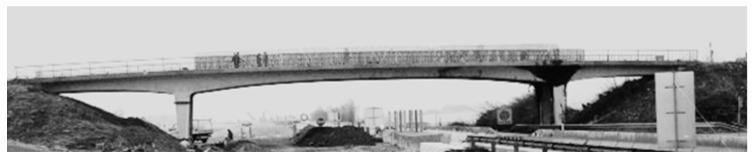
Bridge S101 during testing [145].

**Figure 7 sensors-23-03290-f007:**
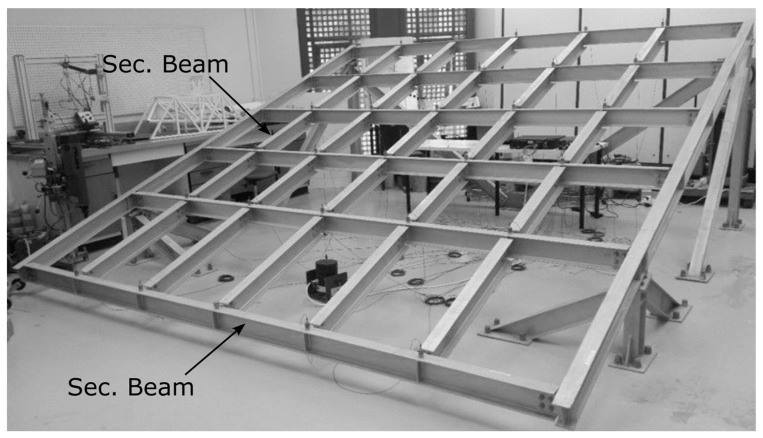
Qatar University grandstand simulator [146].

**Figure 8 sensors-23-03290-f008:**
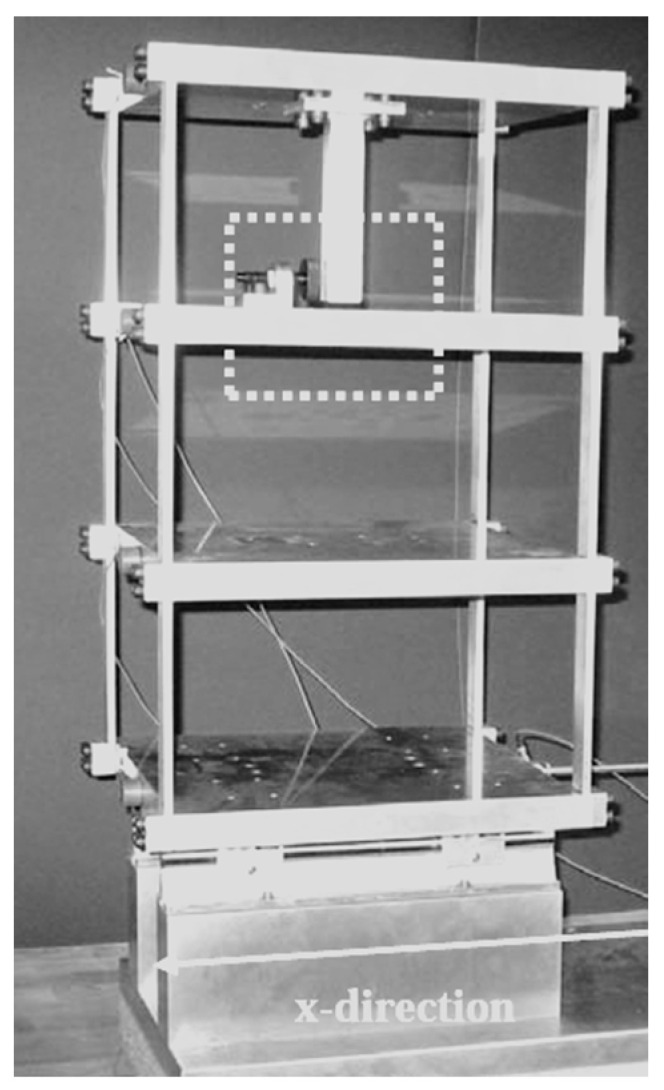
Los Alamos National three-story frame structure [148].

**Figure 9 sensors-23-03290-f009:**
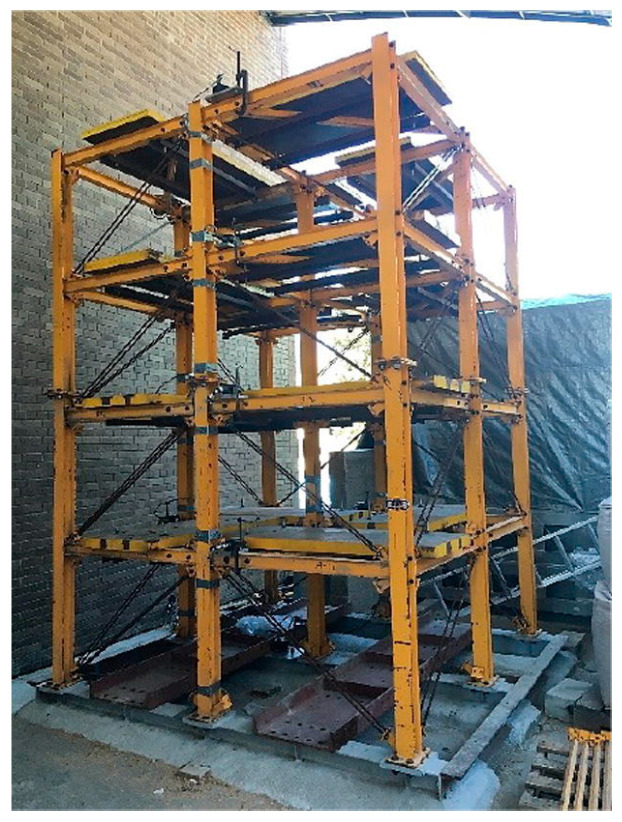
IASC-ASCE benchmark structure [152].

**Figure 10 sensors-23-03290-f010:**
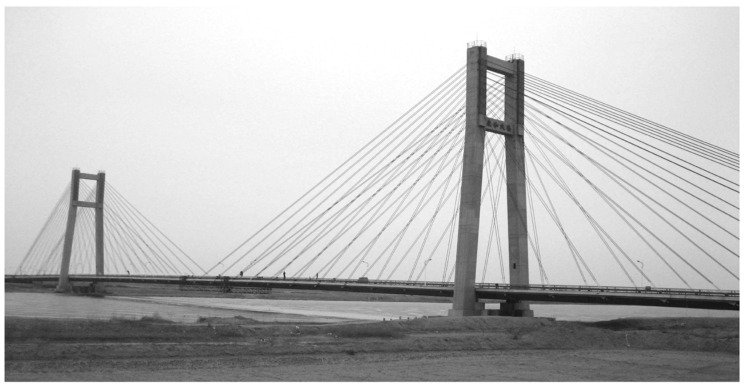
Tianjin Yonghe Bridge [154].

**Figure 11 sensors-23-03290-f011:**
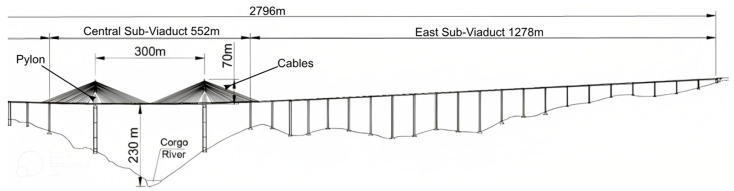
Schematic view of the Corgo Bridge [96].

**Figure 12 sensors-23-03290-f012:**
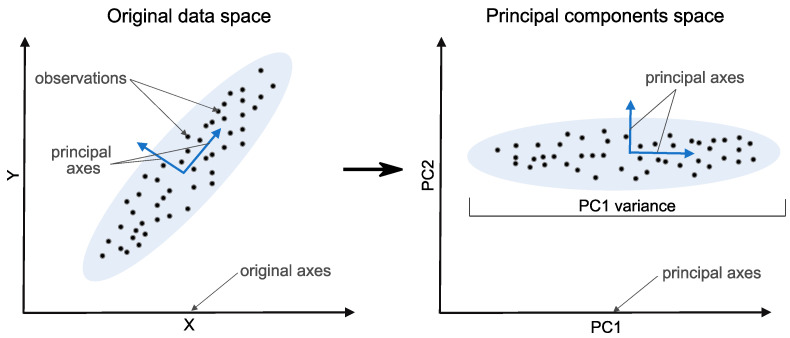
Principal component analysis.

**Figure 13 sensors-23-03290-f013:**
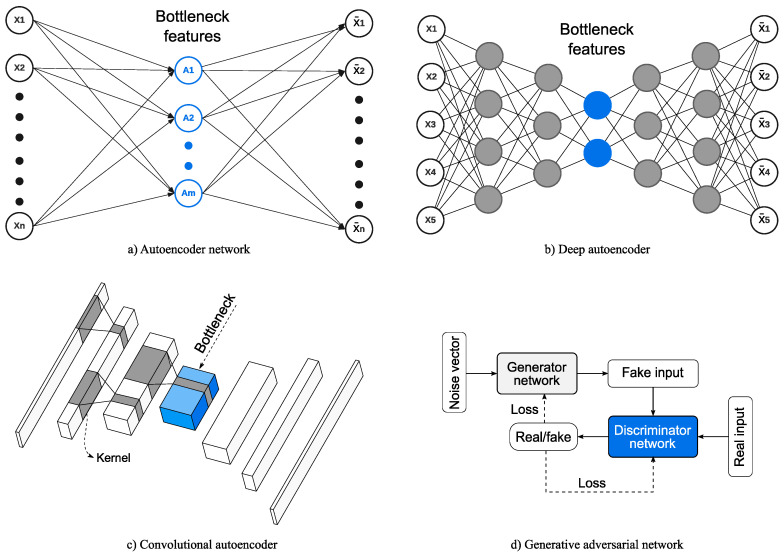
Examples of neural networks used in unsupervised SHM methods.

**Figure 14 sensors-23-03290-f014:**
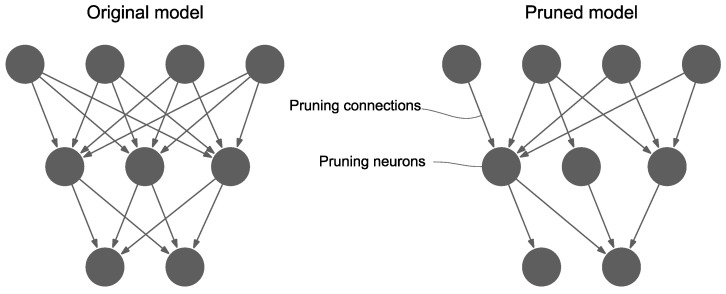
Model pruning.

**Figure 15 sensors-23-03290-f015:**
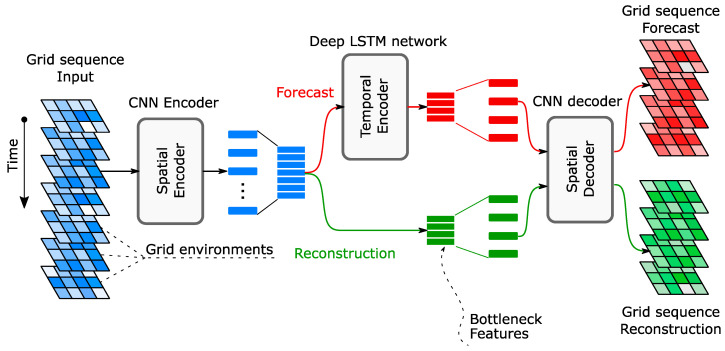
The spatiotemporal composite autoencoder network [58].

**Figure 16 sensors-23-03290-f016:**
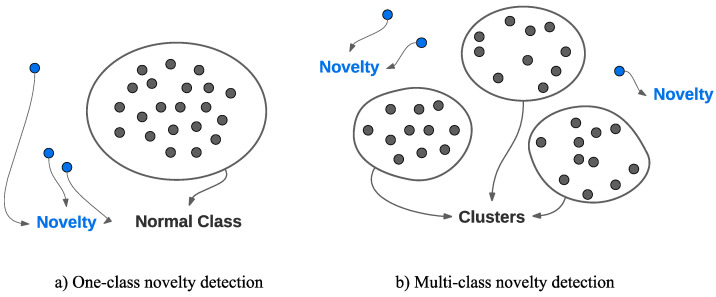
One-class and multi-class novelty detection.

**Figure 17 sensors-23-03290-f017:**
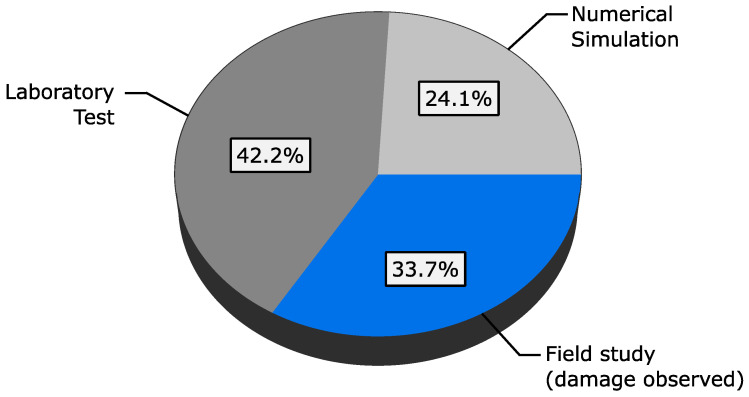
Percentage of the reviewed papers using datasets from field studies, laboratory tests, and numerical simulations. If a paper involves multiple types of datasets, it counts towards field studies, if it has one. Otherwise, it counts towards laboratory tests.

**Table 1 sensors-23-03290-t001:** Applications of unsupervised learning, vibration-based SHM (List of abbreviations is provided in Table 2).

Authors	Year	Feature Extraction	Classifier	Test Structure
Eltouny and Liang [58]	2023	CNN-LSTM hybrid	EVT-based test	Numerical multi-story multi-bay structure
Li et al. [59]	2023	Power cepstral coefficients, GAE	MSD-based outlier analysis	Z-24 Bridge, numerical 8 DOF model
Sadeghi et al. [60]	2023	VMD	-	Laboratory-scale bridge
Soleimani-Babakamali et al. [61]	2023	FFT	GAN (CNN, LSTM)	QUGS, IASC-ASCE benchmark
Entezami et al. [62]	2022	Empirical machine learning	EVT-based test	Z-24 bridge, Tianjin Yonghe bridge
Fernandez-Navamuel et al. [63]	2022	AE-PCA hybrid	Percentile	Beltran bridge, Infante Dom Henrique bridge
Giglioni et al. [64]	2022	AE, ensemble learning	Percentile	Z-24 Bridge
Kim and Song [65]	2022	Flexibility matrices, CVAE	-	Numerical steel structure
Lucà et al. [66]	2022	Modal frequencies	GMM	Tie-rods
Meixedo et al. [67]	2022	ARX, PCA	k-means	Sado Bridge
Shi et al. [68]	2022	ANN, CNN	SVDD	IASC–ASCE benchmark (numerical), three-story masonry frame
Soleimani-Babakamali et al. [69]	2022	FFT	GAN (CNN, LSTM)	QUGS, IASC-ASCE benchmark
Sony and Sadhu [70]	2022	Multivariate EMD	Significance test	Z-24 bridge, numerical 10-DOF model
Wang et al. [71]	2022	NExT, sparse Bayesian learning	Bayesian hypothesis test	Tianjin Yonghe Bridge
Yan et al. [72]	2022	Transmissibility	KL divergence, Bayesian inference	S101 bridge, lab beams, numerical 10-story building
Zhang et al. [73]	2022	WT, Convolutional VAE	Significance test	Laboratory-scale tunnel model
Eltouny and Liang [74]	2021	CIM, PCA	Bayesian-optimized KDME, EVT-based test	Numerical RC frame, numerical high-rise structure, three-story masonry frame
Jiang et al. [75]	2021	AE	Predefined threshold	QUGS, LANL three-story structure
Li et al. [76]	2021	CNN-GCN hybrid	Significance test	Cable-stayed bridge
Ma et al. [77]	2021	PPCA	Q-statistic and T^2^-statistic anomaly detection	Numerical auditorium
Mao et al. [78]	2021	GAF, GAN, CAE	CUSUM	Cable-stayed bridge
Mousavi et al. [79]	2021	VMD	-	Numerical beam
Movsessian et al. [80]	2021	MD, ANN	ROC	Wind turbine
Sarmadi and Yuen [81]	2021	KNFST	EVT-based test	Z-24 bridge, Wooden bridge
Sarmadi et al. [82]	2021	Sequential ensemble (UMD, MSD, local MSD)	EVT-based test	Z24 bridge, wooden bridge
Silva et al. [83]	2021	Stacked AE	GMM	Z-24 bridge
Son et al. [84]	2021	LSTM	Significance test	Cable-stayed bridge
Wang and Cha [85]	2021	AE	OCSVM	Laboratory-scale steel bridge, numerical shelf structure
Yuan et al. [86]	2021	CVAE	Elliptic envelope, OCSVM	Laboratory vehicle-track, numerical vehicle-track
Rastin et al. [87]	2021	CAE	Significance test	Tianjin Yonghe Bridge, numerical IASC-ASCE benchmark, numerical grid structure
Entezami et al. [88]	2020	ARMA	ESD-PKLD hybrid with nearest neighbor	Tianjin Yonghe Bridge
Entezami et al. [89]	2020	ARX	MSD-PKLD hybrid	Tianjin Yonghe Bridge
Entezami et al. [90]	2020	ARMA, AE	MD, EVT-based	Tianjin Yonghe Bridge
Ma et al. [91]	2020	CVAE	-	Laboratory steel bridge
Mousavi et al. [92]	2020	HHT, ANN	-	Laboratory steel truss bridge
Ni et al. [93]	2020	-	CNN	Suspension bridge
Nie et al. [94]	2020	Fixed MPCA	-	Suspension bridge, laboratory beam, numerical beam
Soman [95]	2020	EEMD	POD analysis	Laboratory offshore tripod
Tomé et al. [96]	2020	Johansen test	Hotelling T^2^ control charts	Numerical Corgo Viaduct
Tran et al. [97]	2020	SRIM	Hierarchical clustering, univariate outlier analysis	Laboratory bridge, Steel pedestrian bridge
Xu et al. [98]	2020	WT	EVT-based	Numerical Xihoumen suspension bridge
Bull et al. [99]	2019	Transmissibility	MSD-based outlier ensemble	Z-24 bridge, Gnat aircraft
de Almeida Cardoso et al. [100]	2019	TF-IQRM	k-medoids, student’s *t*-test	IASC-ASCE benchmark, PI-57 bridge
Entezami and Shariatmadar [101]	2019	EEMD, AR, ARX, DTW, PCA	Hotelling T^2^ control charts, QRE, significance test	IASC-ASCE benchmark
Entezami et al. [102]	2019	AR	PKLD, significance test	LANL three-story structure, wooden bridge
Han et al. [103]	2019	TKEO, CEEMD	-	Scaled wind turbine
Ozdagli and Koutsoukos [104]	2019	NExT/ERA, PCA, AE	Euclidean distance	LANL three-story structure, numerical beam
Sousa Tomé et al. [105]	2019	Multilinear regression, PCA	Hotelling T^2^ control charts	Numerical Corgo Viaduct
Anaissi et al. [106]	2018	CANDECOMP/PARAFAC Tensor Decomposition	OCSVM	Cable-stayed bridge, laboratory replica of SHB jack arch
Cha and Wang [107]	2018	CWT, crest factor	density peaks-based fast clustering	Laboratory-scale steel structure
Entezami and Shariatmadar [108]	2018	AR	PAC, RRC, significance test	LANL three-story structure, IASC-ASCE benchmark
Rafiei and Adeli [109]	2018	SWT, FFT	Deep RBM	Laboratory-scale 38-story concrete building
Vamvoudakis-Stefanou et al. [110]	2018	MM learning (PCA, AR)	KL-divergence	Composite beams
Zhou et al. [111]	2018	Transmissibility, PCA	-	IASC-ASCE benchmark, numerical beam
Alamdari et al. [112]	2017	Spectral moments	k-means−−	SHB
Gres et al. [113]	2017	Hankel matrix, MD	Significance test, Hotelling T^2^ control charts	S101 bridge, numerical offshore support structure
Gu et al. [114]	2017	AE w/temperature input	Euclidean distance	Steel grid structure
Langone et al. [115]	2017	Modal frequencies	Adaptive KSC	Z-24 bridge
Neves et al. [116]	2017	ANN	MD, GP	Numerical bridge
Santos et al. [117]	2017	Modal frequencies	GA-EM-GMM	Z-24 bridge
Xia et al. [118]	2017	EEMD	-	Jiangyin suspension Bridge
Zhou et al. [119]	2017	Transmissibility	Hierarchical clustering	Numerical 10-story structure, Laboratory beam
Amezquita-Sanchez and Adeli [120]	2016	SWT, FD	Significance test	Laboratory-scale 38-story concrete building
Avci and Abdeljaber [121]	2016	SOM	-	IASC-ASCE benchmark
Diez et al. [122]	2016	FFT	KNN, k-means	SHB
Mohammadi Ghazi and Büyüköztürk [123]	2016	HHT	MSD-based outlier analysis	Laboratory steel structure
Santos et al. [124]	2016	ANN	k-means, Gowda–Diday dissimilarity	Numerical Guadiana International Bridge
Silva et al. [125]	2016	Modal frequencies	GA-clustering	Z-24 Bridge, Tamar Bridge
Tibaduiza et al. [126]	2016	PCA	T^2^-statistic, Q-statistic, combined index, I^2^ index	Wind turbine blade
Ulriksen and Damkilde [127]	2016	CWT, GDTKEO	MSD-based outlier analysis	Numerical beam, Wind turbine blade
Alves et al. [128]	2015	Symbolic analysis	Dynamic clouds, FCM, hierarchical clustering	Laboratory steel beam, PI-57 bridge
Dervilis et al. [129]	2015	LTS	MCD	Z-24 Bridge, Tamar bridge
Shahidi et al. [130]	2015	SVR, CR, AR, ARX	Significance test	Laboratory steel frame
Döhler et al. [131]	2014	Subspace identification	GLR	Numerical mass-spring chain, numerical truss, numerical beam
Figueiredo et al. [132]	2014	Modal frequencies	MCMC-GMM, MSD-based outlier analysis	Z-24 Bridge
Nigro et al. [133]	2014	Time series modeling, CUSUM, EWMA, MSE	Bootstrapping, Fisher Criterion- MSD significance test	Laboratory steel frame
Figueiredo and Cross [134]	2013	GMM	MSD-based outlier analysis	Z-24 Bridge
Kunwar et al. [135]	2013	HHT	-	Laboratory bridge
Laory et al. [136]	2013	MPCA	Significance test	The Ricciolo viaduct, numerical concrete frame
Sankararaman and Mahadevan [137]	2013	Bond graph model residuals	Bayesian hypothesis test	Numerical frame
Yu et al. [138]	2013	FRF, PCA, KPCA	FCM	Experimental truss bridge
Kesavan and Kiremidjian [139]	2012	WT, PCA	k-means	IASC-ASCE Benchmark (numerical)
Meredith et al. [140]	2012	EMD	-	Numerical Euler–Bernoulli beam

**Table 2 sensors-23-03290-t002:** List of Abbreviations.

Abbreviation	Definition	Abbreviation	Definition
AE	Autoencoder	KNFST	Kernel Null Foley–Sammon Transform
ANN	Artificial Neural Network	KNN	K-Nearest Neighbor
ARMA	Autoregressive Moving Average	KPCA	Kernel Principal Component Analysis
ARX	Autoregressive-Exogenous	KSC	Kernel Spectral Clustering
ASCE	American Society of Civil Engineering	LANL	Los Alamos National Laboratory
CAE	Convolutional Autoencoder	LSTM	Long Short-Term Memory
CEEMD	Complementary Ensemble Empirical Mode Decomposition	LTS	Least Trimmed Squares
CIM	Cumulative Intensity Measure	MCD	Minimum Covariance Determinant
CNN	Convolutional Neural Network	MCMC	Monte Carlo Markov Chain
CR	Collinear Regression	MD	Mahalanobis Distance
CUSUM	Cumulative Sum	MM	Multiple-Model
CVAE	Convolutional Variational Autoencoder	MPCA	Moving Principal Component Analysis
CWT	Continuous Wavelet Transform	MSD	Mahalanobis Square Distance
DTW	Dynamic Time Warping	NExT	Natural Excitation Technique
EEMD	Ensemble Empirical Mode Decomposition	OCSVM	One-Class Support Vector Machine
EM	Expected-Maximization	PAC	Parametric Assurance Criterion
EMD	Empirical Mode Decomposition	PCA	Principal Component Analysis
ESD	Euclidean Square Distance	PKLD	Partition-Based Kullback–Leibler Divergence
EVT	Extreme Value Theory	POD	Probability of Detection
EWMA	Exponentially Weighted Moving Average	PPCA	Probabilistic Principal Component Analysis
FCM	Fuzzy C-Means	QRE	Q-Reconstruction Error
FD	Fractal Dimension	QUGS	Qatar University Grandstand Simulator
FFT	Fast Fourier Transform	RBM	Restricted Boltzmann Machine
FRF	Frequency Response Function	RC	Reinforced Concrete
GA	Genetic Algorithm	RRC	Residual Reliability Criterion
GAE	Generalized Autoencoder	SHB	Sydney Harbor Bridge
GAF	Gramian Angular Field	SOM	Self-Organizing Maps
GAN	Generative Adversarial Network	SRIM	System Realization Using Information Matrix
GCN	Graph Convolutional Networks	SVDD	Support Vector Data Description
GDTKEO	Generalized Discrete Teager–Kaiser Energy Operator	SVR	Single-Variate Regression
GLR	Generalized Likelihood Ratio	SWT	Synchrosqueezed Wavelet Transform
GMM	Gaussian Mixture Models	TF-IQRM	Time–Frequency Interquartile Range-Median
GP	Gaussian Process	TKEO	Teager–Kaiser Energy Operator
HHT	Hilbert–Huang Transform	UMD	Univariate Mahalanobis Distance
IASC	International Association For Structural Control	VAE	Variational Autoencoder
KDME	Kernel Density Maximum Entropy	VMD	Variational Mode Decomposition
KL	Kullback–Leibler	WT	Wavelet Transform

## Data Availability

No new data were created or analyzed in this study. Data sharing is not applicable to this article.

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
