# Peer review of "Unsupervised Learning Methods for Data-Driven Vibration-Based Structural Health Monitoring: A Review"

_sensors, 2023, doi:10.3390/s23063290_

Round 1
Reviewer 1 Report
The present manuscript titled, ‘Unsupervised Learning Methods for Data-Driven Civil Structural Health Monitoring: A Review’ presents a review of the various structural damage detection approaches especially using unsupervised machine learning methods. In the present study, the authors have reviewed articles on vibration-based Structural Health Monitoring methods including field studies, numerical simulations, and Laboratory tests. Subsequently, a detailed account of the feature extraction techniques, deep learning methods, novelty detection, and cluster analysis have been used to detect and predict the extent of damage in the structures being monitored.
In this article, it appears that there is a focus on only vibration-based Civil Structural Health Monitoring, this should be reflected in the title of the article. There are other methods of Civil Structural Health Monitoring paradigms like Ultrasonics, thermographic techniques, Acoustic Emissions and Electrochemical methods. In the introduction, the authors should consider the inclusion of a discussion about the overall applicability of these methods for Civil Structural Health Monitoring for the benefit of the readers. A minor editing issue in Table 2 with regard to the presentation of Abbreviations needs to be checked. For example, CIM should be defined as a Cumulative Intensity Measure. Overall, the article is well-written and can be accepted for publication once the suggested queries have satisfactorily been addressed in the revised version of this manuscript.
Reviewer 2 Report
The authors try to peform a review on a subject that is very wide and where in the recent years have been a huge number of contributions, not only because of the large effort in new tools for unsupervised machine learning, but also because focusing on civil engineering, which is a large field of activity. As a result, I have found a lot of interesting references in the subject. However, I am also missing many others with equal or even more interest. Also, taking into account the actual large number of journals devoted to the subject, I find that very low number of journals were scanned. In my opinion the review is not exhaustive and, in fact, as I have mentioned above it is really difficult to perform it. The authors do not explain why some references are included and others, not. Therefore, they should explain the criteria followed in the state of the art review process. For instance, which journals were selected and others no and why, between which years, which keywords or other selection criteria were used ? The same can be of application to the benchmarks provided. Not all available benchmarks are included. Which is the criteria to include ones and no others ?
After doing that, apart of making a list of references and short explanations of their content, the important in a review paper are the conclusions that can be obtained at the end of the analysis. In this sense, the chapter on challenges and future trends is very weak, in particular, the section on environmental and operational variability. I can not see any conclusion pointing to future challenges and developments on the subject.
Apart from that, these are other specific comments:
1.- The description of the Sydney Harbour bridge is very incomplete. It is impossible to see where and how the SHM was done. This needs a better description including additional figures as the actual text is not enough
2.- In the time-frequency domain, nothing is said about the empirical mode decomposition and variational mode decompostion largely used in the field. Very little references are reviwed on this subject, despite a huge number of numerical, laboratory and in-field realizations available in the literature
Reviewer 3 Report
This review paper presents a comprehensive review for unsupervised learning methods for civil structural health monitoring. The popular experimental datasets, feature extraction techniques, unsupervised learning methods, and deep learning methods are reviewed. The current challenges and some future trends are also discussed. I think readers in Sensors will be very interested in this review paper. Some minor suggestions are given blow for improving the overall quality of this paper.
1. Despite the authors provide a detailed introduction and discussion on unsupervised learning methods and feature extraction techniques, some readers may still not be clear how the damage state of structure can be assessed using these advanced methods. A schematic diagram of structural condition assessment based on unsupervised learning and SHM methods will be helpful for readers’ better understanding.
2. Some popular signal processing methods are reviewed in section 3.2. As the authors stated, “the time-frequency domain representations are currently gaining more attention from SHM researchers.” However, some state-of-the-art time-frequency processing methods are neglected, such as EMD, EEMD, and VMD. Following papers reported the application of these methods in feature extraction and fault diagnosis in materials and structures. I suggest the authors cite them and provide a brief introduction in section 3.2.
http://dx.doi.org/10.1016/j.ymssp.2012.09.015
https://doi.org/10.1088/1361-6501/ac4ed7
https://doi.org/10.1016/j.measurement.2019.02.053
Reviewer 4 Report
The paper is a comprehensive review of the subject matter, with up-to-date references and a concise description of the methods of the main unsupervised learning approaches.
Table 1 should be revised because it contains some minor errors.
In addition, some figures should be improved to match the quality.
The authors talk about unsupervised, but they speak about autoencoders, CNNs, and LSTMs that are trained and supervised.
The interference is unsupervised, but this point needs to be discussed further.
Perhaps the title can be generalized to Learning-based Methods for Data-Driven Civil Structural Health Monitoring: A Review
Round 2
Reviewer 2 Report
The authors have addressed my comments